

# The subtleties of three-dimensional radiative effects in contrails and cirrus clouds

Julie Carles[1,2], Nicolas Bellouin[1,3], Najda Villefranque[4], and Jean-Louis Dufresne[2]

[1]Institut Pierre-Simon Laplace, Sorbonne Université / CNRS, Paris, France
[2]Laboratoire de Météorologie Dynamique (LMD), Sorbonne Université, ENS, Université PSL, École Polytechnique, Institut Polytechnique de Paris, CNRS, 75005 Paris, France
[3]Department of Meteorology, University of Reading, Reading, UK
[4]Centre National de Recherches Météorologiques, UMR 3589 CNRS, Météo France, Toulouse, France

**Correspondence:** Julie Carles (julie.carles@ipsl.fr)

**Abstract.**

The radiative effect of cirrus, contrails, and contrail cirrus affects the energy budget of the Earth and climate change. Those clouds, and especially contrails, are heterogeneous and their holes and sides exert three-dimensional radiative effects. This study uses the htrdr Monte Carlo radiative transfer code to investigate the sensitivity of the cloud radiative effect (CRE) to the geometrical dimensions and optical depth of optically thin ice clouds (cloud optical depth < 4), with particular emphasis on three-dimensional radiative effects. When the Sun is at zenith, an increase in cloud optical depth causes a linear increase in shortwave (SW) CRE but a saturation of longwave (LW) CRE, causing the net CRE to change sign from positive to negative. The optical depth at which this change in sign occurs depends on the cloud geometry. 3D effects make the one-dimensional SW and LW CREs more positive for a Sun at zenith, reaching the same order of magnitude as the 1D CRE itself for clouds with high aspect ratios. The angular dependence of ice crystal scattering strongly increases shortwave CRE when solar zenith angle increases. 3D effects change sign from positive at zenith to negative at large zenith angles as the Sun's rays interact more with the cloud sides. Integrating instantaneous CRE and 3D effects over selected days of the year indicates compensation of SW with LW 3D effects for some cloud orientations, but 3D effects remain important in some cases. These results suggest that the 3D structure of cirrus and contrails needs to be considered to finely quantify their CRE and radiative forcing.

## 1 Introduction

Cirrus clouds play an important role in the radiative budget of the Earth (Heymsfield et al. (2017)) due to their large global coverage of 30%, with a maximum in the Tropics (Nazaryan et al. (2008)). On a global scale, cirrus clouds have a positive net radiative effect (Heymsfield et al. (2017); Hong et al. (2016); Chen et al. (2000)). In addition to natural cirrus, anthropogenic cirrus also contribute to the radiative budget and climate change. Originating from aircraft flying in high altitude air masses, aviation induced cloudiness (AIC) consists in young, linear shaped condensation trails and persistent contrails that may evolve in cirrus clouds, often indistinguishable from natural cirrus (Kärcher (2018)). Their impact on climate, growing along with the rapid increase in air traffic, is part of the non-$CO_2$ effects of aviation on climate (Lee et al. (2021)). These non-$CO_2$ effects may





account for a significative fraction of aviation effective radiative forcing, with high uncertainty, mostly due to aviation induced cloudiness.

An uncertainty of about 55% in the AIC effective radiative forcing is attributed to the radiative response to contrail cirrus (Lee et al. (2021)). The associated contributions arise from the inhomogeneity of ice clouds and cloud overlap within a grid box, the radiative transfer scheme (Myhre et al. (2009)), and notably from the use of plane parallel radiative transfer instead of full 3D radiative transfer. The three-dimensional effects of radiation have previously been demonstrated to be non negligible in the radiative effect of contrails and cirrus (Gounou and Hogan (2007); Forster et al. (2012)). However, these effects are

overlooked in contrail studies based on climate modelling (Bickel et al. (2020)) and simplified process-models (Teoh et al. (2020); Teoh et al. (2024)) that rely on plane-parallel, two-stream radiative transfer.

This article examines the 3D radiative effects of contrails and cirrus clouds, with a specific focus on the radiative effects of the sides of the clouds rather than those resulting from inhomogeneities in cloud properties. The study explores three key quantities that significantly impact cloud radiative effect: cloud width, cloud height and cloud optical depth (Meerkötter et al.

(1999); Wolf et al. (2023)). By doing so, a wide range of configurations representative of contrails and cirrus clouds can be examined. We present a comprehensive analysis of the cloud radiative effect, focusing on optically thin clouds, and explore a few simple cases integrated over space and time as a first step towards an assessment of the importance of 3D radiative effects at the global scale. In particular, the subtle balance between the longwave and shortwave cloud radiative effects is investigated, along with its implications for the sign of the net cloud radiative effect.

Section 2 describes the selected cloud configurations and radiative transfer tools used. The remainder of the paper is divided into four sections. Section 3 describes the behaviour of CRE for a Sun at zenith, while Section 4 explores the influence of solar zenith and azimuth angles on CRE. Each of these sections studies plane parallel calculations first, before examining 3D effects. Section 5 focuses on the behaviour of 3D radiative effects when integrated over time. Finally, Section 6 discusses and summarises the results.

## 45   2   Method

### 2.1   Representation of an idealised contrail or cirrus

For the study, we use a domain about 60 km wide, i.e. of typical climate model grid box dimensions, containing a homogeneous idealised parallelepiped cloud whose top is located at 11 km of altitude (as in Kärcher (2018); Krämer et al. (2020)). Figure 1 illustrates the cloud and its position relative to the Sun. Its width is denoted $w$ and height $h$. The cross-section of the cloud

is rectangular, differing from previous studies which used an ellipsoidal cloud (Gounou and Hogan (2007) and Forster et al. (2012)). However, this choice does not affect the relevance of the results, as will be demonstrated in section 4.3.

To cover a wide range of cloud properties representative of contrails and cirrus (Kärcher (2018); Krämer et al. (2020)), calculations are performed for various cloud widths ($w$) and heights ($h$), as well as a number of cloud optical depths $\tau_c$, listed in Table 1. Clouds of different height share a common cloud top altitude at 11 km to keep cloud top temperature constant





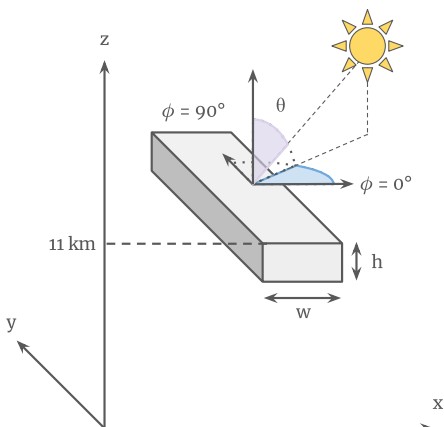

**Figure 1.** Schematic of the idealised cloud used in this study and its position relative to the Sun. Cloud top height is kept constant at 11 km. Its width $w$ varies between 0.5 and 16 km, its thickness $h$ ranges between 0.25 and 2 km with the condition $w \geqslant h$. The solar zenith angle $\theta$ is defined as the angle formed between the vertical axis and the direction of the Sun. The azimuth angle $\phi$ is the angle formed between the x-axis and the Sun direction in the horizontal plane. For an azimuth of $0°$ (respectively, $90°$), the Sun is aligned with the x (resp., y) direction and is perpendicular (resp., parallel) to the cloud.

**Table 1.** Summary of cloud parameters.

| | |
|---|---|
| Cloud width $w$ (km) | 0.5, 1, 2, 4, 8, 16 |
| Cloud geometrical thickness $h$ (km) | 0.25, 0.5, 1, 2 |
| Cloud optical depth at 550 nm $\tau_c$ | 0.03125, 0.0625, 0.125, 0.25, 0.5, 1, 2, 3, 4 |
| Cloud fraction $f$ | $w/L$ |
| Domain averaged ice water path ($aIWP$, kg/m$^2$) | $w/L \times \tau_c/k_{ext}$ |
| | with $k_{ext}$ the extinction coefficient in m$^2$/kg |

through all configurations. To remain in a realistic range of contrail and cirrus geometries, the cloud cannot be thicker than it is wide, i.e. we must have $w \geqslant h$.

## 2.2   Radiative transfer

The radiative transfer (RT) calculations were done with htrdr, a 3D Monte-Carlo radiative transfer code (Villefranque et al. (2019)). The code takes as inputs a 3D field of cloud water concentration with associated optical properties.

The ice crystals optical properties required by htrdr are the absorption and scattering cross sections, and the asymmetry parameter $g$ of the Henyey-Greenstein phase function. The optical properties of individual particles are taken from Yang et al. (2013) and Bi and Yang (2017) (Laurent Labonnote, personal communication). These properties are integrated over a particle size distribution provided by Field et al. (2007), which depends on the cloud ice water content (IWC) and temperature (T).





The result is a look-up table describing the bulk ice optical properties as a function of in cloud IWC and T. Absorption and scattering coefficients are discretized over 274 wavelengths between 0.2 and 2.99 microns for the shortwave (SW) spectral domain, and 171 wavelengths between 3 and 99 microns for longwave (LW). For our study, we considered a single shape, hexagonal column, to represent the cloud microphysics, as in Gounou and Hogan (2007) and Forster et al. (2012). As our focus lies in exploring the sensitivity of the radiative effect of clouds to their geometric properties, optical properties of ice crystals are assumed constant within the cloud and calculated with an ice water content of $1 \times 10^{-5}$ kg/m$^3$ and a temperature of 227.5 K, which are in the range of values considered in the previous section. Therefore, all cloud configurations studied in this paper share identical optical properties of ice crystals.

For the atmosphere, a standard midlatitude summer profile (MLS) is used for temperature, pressure and water vapor. The surface is perfectly absorbing, its albedo is 0 both in the longwave (LW) and shortwave (SW) domain.

The cloud radiative effect (CRE) is calculated as the difference in net fluxes at the tropopause (13 km) between clear sky and all-sky conditions:

$$CRE_{net} = CRE_{LW} + CRE_{SW} = (F_{clear} - F_{all-sky})_{LW} + (F_{clear} - F_{all-sky})_{SW} \qquad (1)$$

with $F$ the upwelling flux at 13 km. A positive CRE indicates a cloud warming the troposphere and surface. Four independant runs are necessary in order to calculate the four terms in Eq. 1 (clear sky and all-sky in both SW and LW spectral domains).

The Monte-Carlo radiative transfer code htrdr evaluates the intensity of radiation reaching every pixel of a rectangular virtual sensor by tracing paths of a large number of photons within each pixel. With the Monte Carlo method, the random error in the radiative flux rendered by the code reduces proportionally with the square root of the sampled number of photons. In order to reach sufficiently low relative errors, typically $10^9$ photons are sampled to calculate fluxes in this study. The resulting errors are usually below $1\%$ percent for calculation of CREs, except in cases where the net CRE or 3D effects are very close to zero, yielding relatively large errors.

As the study focuses on the impact of horizontal transport of radiation, particular attention is needed to take it into account in radiative transfer calculations. Since fluxes are reported far above cloud top, the size of the domain on which radiative transfer is solved needs to be larger than the cloud (in the x direction) to ensure all radiative effects near cloud sides are accounted for. For a Sun at zenith, fluxes are retrieved over a domain with a width of more than 60 km. For large zenith angles, two vertical sensors are added between 11 and 13 km in order to intercept all horizontal radiation and avoid lost photons (see Appendix A). In the y direction, the sensor is placed above the center of the cloud which is then seen as infinite in this direction. Small deviations in the sensor position in this direction do not impact the results.

With the domain of length $L$ and the cloud of width $w$, cloud fraction is proportional to cloud width following the relation $f \propto w/L$. Because radiative transfer is solved on a large domain with $L = 66$ km, the resulting cloud fractions are small: they range between 0.8 and 24%.

In the following sections, a distinction will be done between the all-sky cloud radiative effect (aCRE) and overcast cloud radiative effect (oCRE) with $aCRE = f \times oCRE = (w/L) \times oCRE$. The aCRE represents the cloud's radiative influence on the whole studied domain, taking into account both clear and cloudy areas. It is given in W/m$^2$ averaged over the domain. The



oCRE corresponds to the radiative effect of the cloud only, without the surrounding clear sky areas. The values of oCRE are in W/m$^2$ of cloudy surface area. This notation is also used for other parameters, such as the ice water path: $aIWP = f \times oIWP$, where $oIWP \propto \tau_c$. The $aIWP$ is used to study domain-averaged quantities, while $\tau_c$ focuses on cloudy parts.

The calculations in sections 3.1 and 4.1 are performed with the independent column approximation. To achieve this with htrdr, the size of the horizontal mesh is multiplied by a very large number ($10^6$) as in Barker et al. (2012). This ensures that the magnitude of horizontal transport of radiation between columns becomes negligible compared to vertical fluxes: this is equivalent to treating each cell independently as in plane parallel calculations.

## 3 Sun at zenith

### 3.1 Cloud radiative effect with the plane-parallel approximation

This section provides an overview of the radiative effect of the idealised clouds. A study of the behaviour of the CRE under parallel plane conditions allows a better understanding of the 3D effects later on. We start by examining the SW and LW components of the CRE in turn, before analyzing the net SW+LW CRE.

#### 3.1.1 Shortwave radiative effect

Figure 2 (a) shows the SW aCRE of all cloud configurations as a function of the domain-averaged ice water path (aIWP) for a Sun at zenith. Clouds reflect a larger fraction of solar irradiance back to space than the surface, resulting in a cooling effect for the atmosphere and surface below them. The SW aCRE exhibits a remarkable linear increase with ice content. There is notably very little to no spread between the curves of all cloud configurations, i.e. almost no dependence of the aCRE to cloud geometry.

This behavior can be explained by assuming a linear relationship between cloud reflectance and optical depth for very thin clouds ($\tau_{cloud} << 1$): $R_{cloud} \propto \tau_{cloud}$. The cloud optical depth of an homogeneous cloud is proportional to its ice water path ($\tau_{cloud} \propto \frac{aIWP}{w/L}$). Thus, by multiplying cloud reflectance by cloud fraction (i.e. $w/L$) to get a domain reflectance, one finds that the domain reflectance is directly proportional to the total mass of ice it contains:

$$R_{domain} \propto R_{cloud}\, w/L \propto aIWP \qquad (2)$$

As the SW aCRE is proportional to the domain reflectance ($aCRE_{SW} = \mu_0 F^\downarrow R_{domain}$ with $\mu_0$ the cosine of the solar zenith angle and $F^\downarrow$ the incident solar flux at cloud top at zenith), it inherits the proportionality to the total mass of ice.

The above relationship lies on the assumption of linearity between cloud reflectance and optical depth. This linearity is observed in htrdr for values of $\tau_c$ up to 4 (Fig. 3(a)). We seek to determine whether this behavior can be reproduced using simple expressions, in such a domain of validity in terms of cloud optical depth. There are many theoretical calculations of cloud reflectance in the literature, most of them well reproducing the observed linearity of the CRE with ice content, but often with an overestimated slope (see Fig. C1). An expression that fits the htrdr results very well can be derived from the two-stream approximation, assuming conservative scattering in the SW (i.e. a single scattering albedo $\omega_0 = 1$):



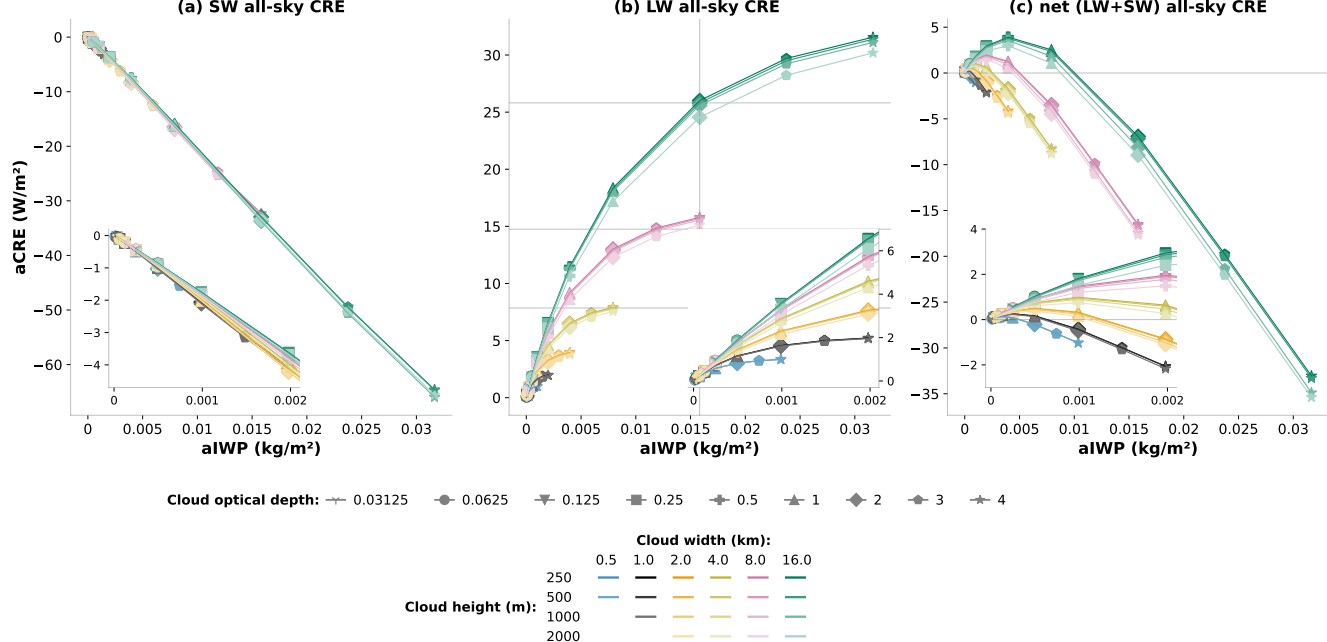

**Figure 2.** All-sky cloud radiative effect (a) in the LW, (b) in the SW and (c) net (SW+LW) as a function of the domain-averaged ice water path, for a Sun at zenith. The colors correspond to cloud width $w$ between 500 m (black) and 16 km (green), and the shades correspond to cloud height $h$ (the darkest shade for clouds 250 meters thick and lightest for 2 km thick cloud, illustrating that the ice water content goes from densest to lightest). The calculations are done with the plane parallel method and the radiative effects are domain-averaged (in $W/m^2$ of the domain). Symbols indicate the cloud optical depth.

$$R_{cloud} = \frac{1}{1 + \gamma_1 \tau_c}[\gamma_1 \tau_c + (\gamma_3 - \gamma_1 \mu_0)(1 - e^{-\tau_c/\mu_0})] \tag{3}$$

when using the «hybrid modified Eddington-delta function» method for the coefficients $\gamma_1$ and $\gamma_3$ (Meador and Weaver (1980), hereafter MW80). The coefficients are function of the asymmetry parameter $g$, the single scattering albedo $\omega_0$ and the cosine of the solar zenith angle $\mu_0$ (see Appendix C1). Taking $\omega_0 = 1$ and $g = 0.8$ (which correspond to optical properties of the ice crystal shapes considered, see Fig. E1 in appendix) a Sun at zenith and the same incoming solar flux at cloud top as htrdr, we obtain that the approximated SW CRE is close to htrdr, except for a slightly overestimated slope (Fig. 3(a)). Figure

C1 in appendix compares this approximation along with several others found in the literature, most of them showing this quasi linearity up to a cloud optical depth of 4, but with very different slopes that do no match htrdr results.

As $\gamma_1$ and $\gamma_3$ do not depend on the cloud optical depth, a Taylor expansion of Eq. 3 gives $R_{cloud} \propto \tau_c$ for $\tau_c << 1$ (say $\tau_c$ up to 0.5). However, this linearity persists for the whole range of studied optical depths, up to $\tau_c = 4$. This might be attributed to the scattering phase function of ice crystals which exhibits a strong peak in the forward direction, adding to the forward

radiation as if scattering had not occured, thereby artificially reducing the «effective» cloud optical depth. Scaling $\tau_c$ by a




factor $(1-g)$ helps accounting for the large fraction of forward scattered radiation (Pierrehumbert (2010)). For the ice crystals considered here ($g \sim 0.8$), the scaled optical depth $\tau_{scaled} = (1-g)\tau_c$ stays below 0.5 up to $\tau_c \simeq 2.5$ and extends the validity range of the linear approximation. It is interesting to note that, as MW80 slightly deviates from linearity above $\tau_c = 2.5$, htrdr results show a very linear relationship for all studied cloud optical depths.

In summary, in the SW, the oCRE increases linearly with cloud optical depth up to 4 and, from a domain point of view, the two opposite effects of increasing cloud fraction and decreasing cloud optical depth when spreading a cloud of fixed ice mass perfectly compensate each other in the aCRE. This results in the SW aCRE being almost independent of cloud geometry for a given mass of ice.

### 3.1.2 Longwave radiative effect

Figure 2(b) shows the LW aCRE of all cloud configurations as a function of the domain-averaged ice water path (aIWP). As expected, the longwave radiative effect of clouds is positive, and for a given cloud geometry, increases with aIWP. A rapid linear increase at small aIWP is followed by a slower increase in all cases. There is also a dependence on cloud geometry, mainly on cloud width and, to a much lesser extent, cloud thickness. For example, considering the three groups of points located at $aIWP = 0.008$ kg/m$^2$, marked by a vertical line, and moving up from one group of points to another, the cloud width increases from 4 (yellow) to 8 (pink) to 16 km (green), and the LW aCRE increases first by more than 60%, followed by another 40% (7.8 to 12.9 to 18.2 W/m$^2$). For a constant aIWP, when spreading a cloud with a given ice mass, cloud fraction increases and cloud optical depth decreases. Separately, these two mechanisms have opposite effects on the aCRE: an increase in cloud fraction leads to an increase in aCRE, while a decrease in cloud optical depth decreases the aCRE. The positive effect on the LW aCRE of spreading the cloud dominates over the negative effect of reduced cloud optical depth.

One can look at the LW aCRE in more detail which, neglecting the effect of scattering by ice crystals and assuming an isothermal cloud, is given by:

$$aCRE_{LW} = f \int_\nu A_\nu [F_\nu^\uparrow - B_\nu(T_{cloud})]d\nu \qquad (4)$$

where $B(T)$ is the Planck irradiance at temperature $T$, $A$ the absorptivity of the cloud and $F^\uparrow$ the upward longwave flux at cloud base. The integral term corresponds to the oCRE and strongly depends on cloud optical depth (Fig. 3(b)). This dependence is expressed through the absorptivity, which can be reproduced with a simple expression by assuming a spectrally constant absorption, neglecting the effect of scattered radiation and using the diffuse approximation:

$$A = 1 - e^{-\tau_c(1-\omega_0)/\bar{\mu}} \qquad (5)$$

The cloud optical depth $\tau_c$ is scaled by a factor $1-\omega_0$, giving the absorption optical depth. A value of $\omega_0 = 0.6$ is taken here (cf appendix), in accordance with the single scattering properties of ice crystals. $\bar{\mu}$ is the diffusivity factor, 0.6 (Elsasser (1942)).

The approximated LW oCRE is close to that obtained with htrdr: it captures the strong increase at low optical depths followed by the weakened slope, caused by the exponential term in cloud absorption (Fig. 3(b)). At optical depths larger than 4 (not shown in the figure), the slope decreases further and the LW oCRE saturates.





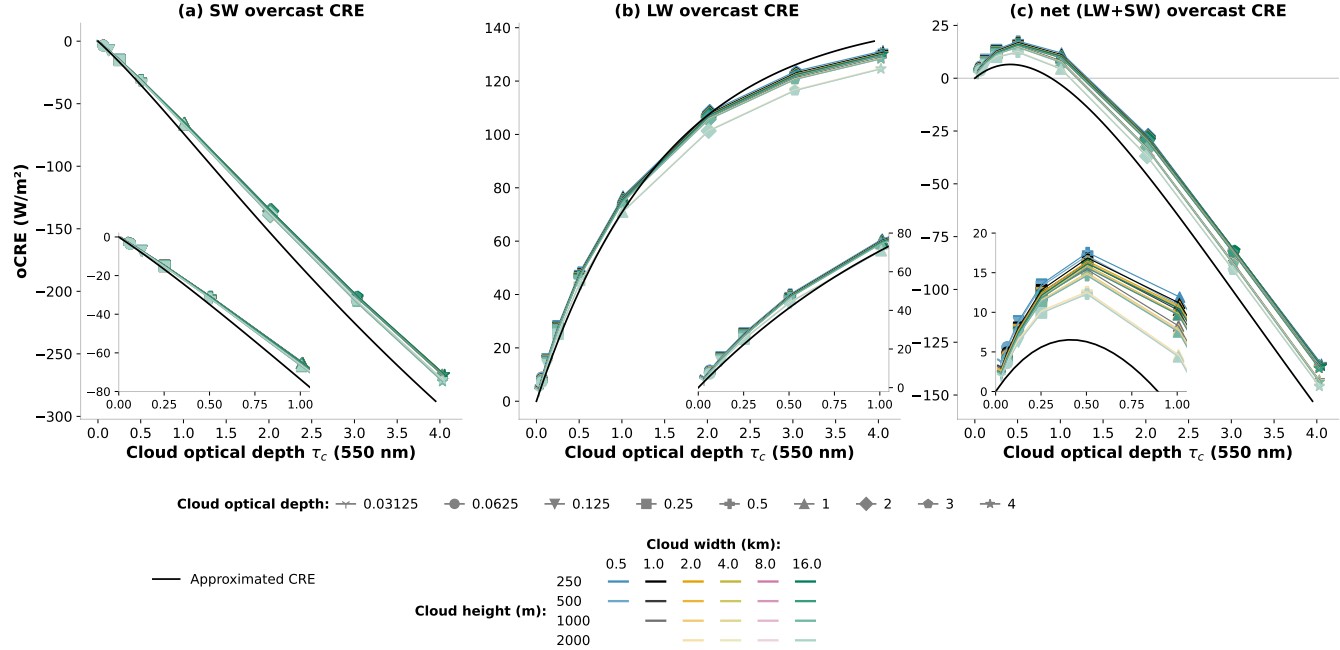

**Figure 3.** Overcast cloud radiative effect (a) in the LW, (b) in the SW and (c) net (SW+LW) as a function of cloud optical depth. Calculations done with htrdr using the plane parallel method are in identical colors as in Fig. 2. Compared with Fig. 2, all curves overlap because the oCRE is independent of cloud width. Symbols indicate cloud optical depth. The oCRE approximated with the simple model is plotted in black lines.

At low optical depths, the cloud absorptivity (and therefore the oCRE) varies linearly with the cloud optical depth, leading to the same linearity with aIWP in the LW aCRE as observed in the SW aCRE (Fig. B1), and for identical reason, but for a

175 shorter range of optical depths below $0.5$. In the LW, the optical depth is scaled with $(1 - \omega_0)$ to remove scattering from the absorption term. The difference between SW and LW lies in the difference in «effective» cloud optical depth, materialized by the terms in the exponential in the expressions for reflectance and absorption. In the SW, $(1 - g) \sim 0.2$, whereas in the LW, $(1 - \omega_0)/\bar{\mu} \sim 0.67$. This ends up in different limit behaviors between the two spectral domains. In general, SW CRE is more complex than LW CRE, which is easier to approximate and analyse.

The LW radiative effect slightly depends on the vertical extent of the cloud. In Fig. 2 (b), dark curves are associated with larger aCREs than lighter ones, indicating that the LW aCRE decreases with increasing vertical extension. This effect can be explained by the dependence of longwave emission on temperature: two clouds sharing cloud top height and optical depth but with different base altitudes cover different ranges of atmospheric temperature profiles. As temperature decreases with altitude in the troposphere, the cloud with the lower base is distributed over warmer temperatures, therefore emitting more radiation

than a cloud with a higher cloud base and reducing the LW aCRE.





### 3.1.3 Net radiative effect

The behavior of the net CRE is driven by the subtle balance between the SW and LW CRE, making estimation of the sign of instantaneous radiative effect of contrails or cirrus clouds sensitive to small numerical differences. As detailed previously, both SW and LW aCRE increase linearly at low ice contents, with the LW aCRE showing a steeper slope than the SW. For higher ice contents, the slope of the LW aCRE starts decreasing while the SW aCRE remains linear.

Despite the SW and LW aCRE being close to linear for low optical depths, as they nearly cancel each other, small non-linearities add up and their linearity is not completely reflected in the net aCRE (this is visible in Fig. B1 which is equivalent to Fig. 2 with only the points where $\tau_c < 1$). As a result, the net aCRE is initially positive and increasing, and then starts to decrease and eventually change sign to become negative at higher contents (Fig. 2(c)).

In addition to its dependence in aIWP, cloud geometry also plays an important role in the net aCRE. For a given aIWP, a progressive horizontal spreading of clouds leads to an enhanced warming or reduced cooling, depending on the sign of the CRE (Fig. 2(c)), following the behavior of the LW aCRE. Within a critical range of aIWP, the sign of the aCRE becomes dependent on the characteristics of the cloud, i.e. two clouds with identical mass of ice can exert either a positive or a negative radiative effect on the domain, depending on their horizontal geometry. Vertical extension plays a role via the LW contribution but remains weak compared to the other effects.

Considering now the oCRE, cloud width no longer comes into play and the analysis can be done with respect to cloud optical depth only. The analytical expressions for SW and LW oCRE introduced in the previous sections clearly demonstrate the general evolution of the net oCRE, illustrating the transition from positive values at low cloud optical depths to negative values at higher cloud optical depths where the LW oCRE saturates and the cooling effect of the SW becomes dominant (Fig. 3(c)). However, the overestimation of the magnitude of the SW CRE, which appeared to be relatively low when considered in isolation, introduces a significant error in the net CRE at low cloud optical depths, enhanced by the weak values taken by the net CRE.

To summarize, the aCRE depends mainly on the aIWP as long as cloud optical depth remains low. When this is no longer the case, cloud fraction comes into play. The aCRE changes from a positive value for low values of aIWP to negative values for higher values of aIWP. The value at which the sign changes depends strongly on cloud fraction.

## 3.2 Three-dimensional radiative effects at zenith

The previous section gave an overview of the radiative effect of an idealised cloud calculated using the plane parallel approximation. Three-dimensional effects of radiation were intentionally ignored. Under the plane parallel approximation, no radiation is allowed to travel horizontally between clear and cloudy sky: photons can only escape the cloud through its base or top. In other words, the impact of the cloud sides on photon transport is neglected. From this section on, horizontal transport of radiation is taken into account in the simulations, allowing to compare against plane parallel results and study the influence of these effects on the CRE.





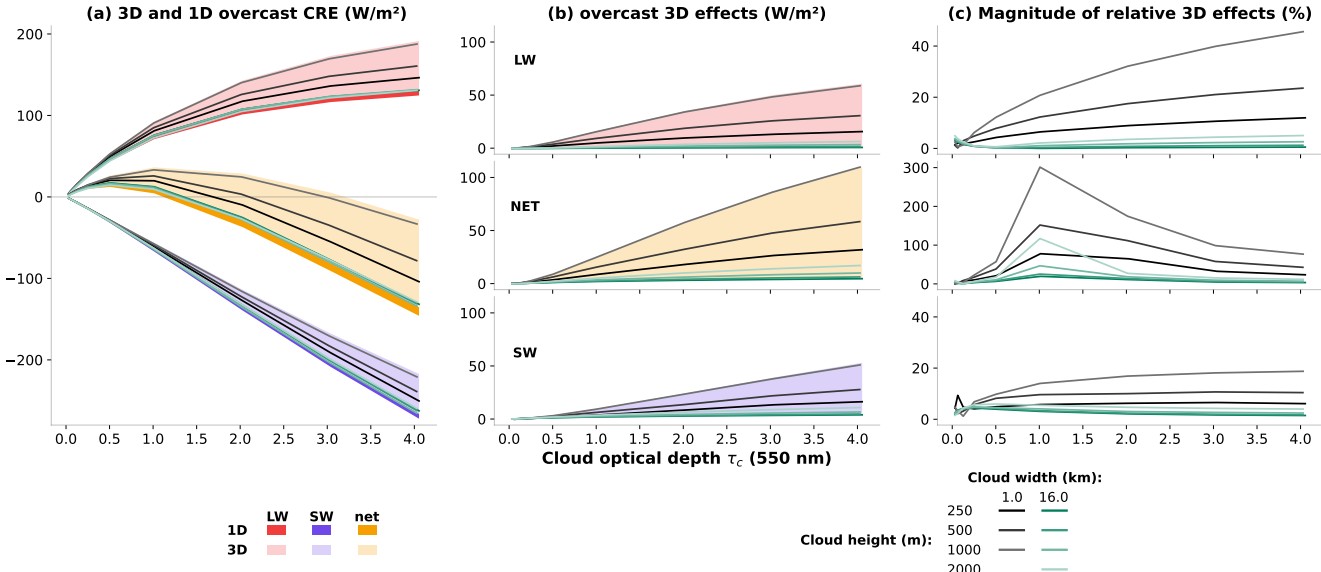

**Figure 4.** (a) Overcast cloud radiative effect, in W/m$^2$, as a function of cloud optical depth. Dark lines represent 1D calculations, light shadings show the range of 3D calculations for various cloud heights and widths. Red is for LW, purple for SW and orange for net (SW+LW) CRE. Blue and green lines correspond to two cloud configurations, with 1 km and 16 km width, respectively. They both come in different shades, from dark to light, to represent the cloud height h, between 250 m and 2 km (b) 3D effects, i.e. difference between 3D and 1D oCREs in W/m$^2$ and (c) relative difference between 3D and 1D calculations, for each spectral domain : LW on top, SW on the bottom row and net in the middle.

Figure 4(a) shows the oCRE of the different cloud configurations, for 1D and 3D calculations, for a Sun at zenith. Accounting for 3D effects in simulations does not modify the qualitative behavior of the oCRE. However, in both spectral domains, the

range of 3D calculations is more positive than that of the 1D, indicating that the 3D effects, i.e. the difference between 3D and 1D CREs, are positive (heating effect). In the LW, the warming effect of clouds is enhanced by 3D effects: unlike in plane parallel calculations, radiation emitted by the surface and atmosphere below the cloud can be absorbed by cloud sides. Cloud sides also become sources of emission: they emit at cloud temperature, i.e. less energy than the surrounding upwelling radiation coming from the lower and warmer atmosphere and surface. These additional sources of absorption and emission modify the

CRE by reducing the outgoing LW radiation. In the SW with the Sun at zenith, incident solar radiation entering the cloud can be scattered through the sides towards the surface. In 1D, these photons would stay within the cloud and therefore have a higher probability of being sent back to space after one or several scattering events. This results in a decreased proportion of reflected radiation in the 3D calculations, equivalent to a reduction in the SW upward flux. Hence, for a Sun at zenith, 3D effects lead to a reduction of the upwelling radiation above the cloud in both spectral domains, adding up to a positive, warming component

to net CRE.





Whereas the oCRE is well understood in plane parallel calculations, 3D calculations add complexity and extra degrees of freedom. Hence, the range of values taken by the oCRE in 3D calculations is very large compared to the 1D calculations. For instance, the effect of cloud geometrical thickness on 3D effects is illustrated in Fig. 4: for all cloud widths and cloud optical depths, geometrically thicker clouds exert larger 3D effects.

It is interesting to note that the effect of cloud height is opposite in 1D and 3D computations of the LW oCRE. We previously showed that for 1D calculations, geometrically thick clouds have a weaker LW oCRE than thin clouds because of their lower cloud base and therefore warmer temperatures. In 3D, as the cloud vertically spreads, the surface of its sides increases, thus enhancing 3D effects and therefore the LW oCRE. For the configurations studied here, this behavior dominates the opposing 1D effect and the overall effect of cloud height is positive in 3D.

This effect of height does not contribute equally to CREs of all clouds geometries: it can double the CRE of narrow clouds (black lines, 1 km width), whereas it has very limited impact on the CRE of wide clouds (green lines, 16 km width). This is because cloud height needs to be compared to its width. The aspect ratio (geometrical thickness to width) of the 16 km cloud is very small. The contribution of cloud sides in the CRE is thus small relative to its wide surface. As a result, the magnitude of 3D effects becomes negligible relative to the net oCRE. When considering the influence of the cloud on the whole domain,

i.e. the aCRE instead of the oCRE, we obtain from empirical analysis that in the LW, 3D effects are nearly insensitive to cloud width and they depend almost linearly on the product of cloud optical depth and height (Fig. 5(b)). As cloud optical depth is proportional to aIWP-to-cloud-width ratio, we can establish that 3D effects in the LW increase linearly with the product of aIWP, cloud width and height ($CRE_{LW}^{3D} \propto h\, aIWP/w$). This relationship works for the LW spectrum but it appears that those parameters are not sufficient to determine the behavior of 3D effects in the SW, at least for extreme cases (Fig. 5(a)).

Thus, when looking at the aCRE, 3D effects of radiation are dependent on cloud height, and indirectly on cloud width via the optical depth (because $\tau_c \propto aIWP/w$). When considering the oCRE, the cloud width is introduced directly in the equation ($oCRE \propto aCRE/w$), and hence introducing the aspect ratio indirectly.

Three-dimensional effects can have a great impact on the net CRE. For optically thin clouds which have a net CRE close to 0 under the plane parallel approximation, 3D effects can be of the same magnitude as the 1D CRE itself. For some cloud

optical depths, two cloud geometries can exhibit net oCRE of opposite signs because of 3D effects (Fig. 4 (a)).

## 4    Dependence on solar zenith and azimuth angles

### 4.1    Plane parallel approximation

The sensitivity of the CRE to solar zenith and azimuth angle is now investigated. For the sake of simplicity, we choose a fixed cloud configuration with a width of 1 km and height 0.5 km. Three cloud optical depths of 0.25, 1 and 4 are selected to pursue

our prior analysis on the behavior of the CRE of optically thin clouds. Solar zenith angle (SZA) is varied between 0 and 88 degrees, and solar azimutal angle between 0 and 90 degrees. Cloud dimensions are similar that of Gounou and Hogan (2007) to allow comparison with their results. The LW CRE has no dependence on SZA as the temperature profile is kept constant.



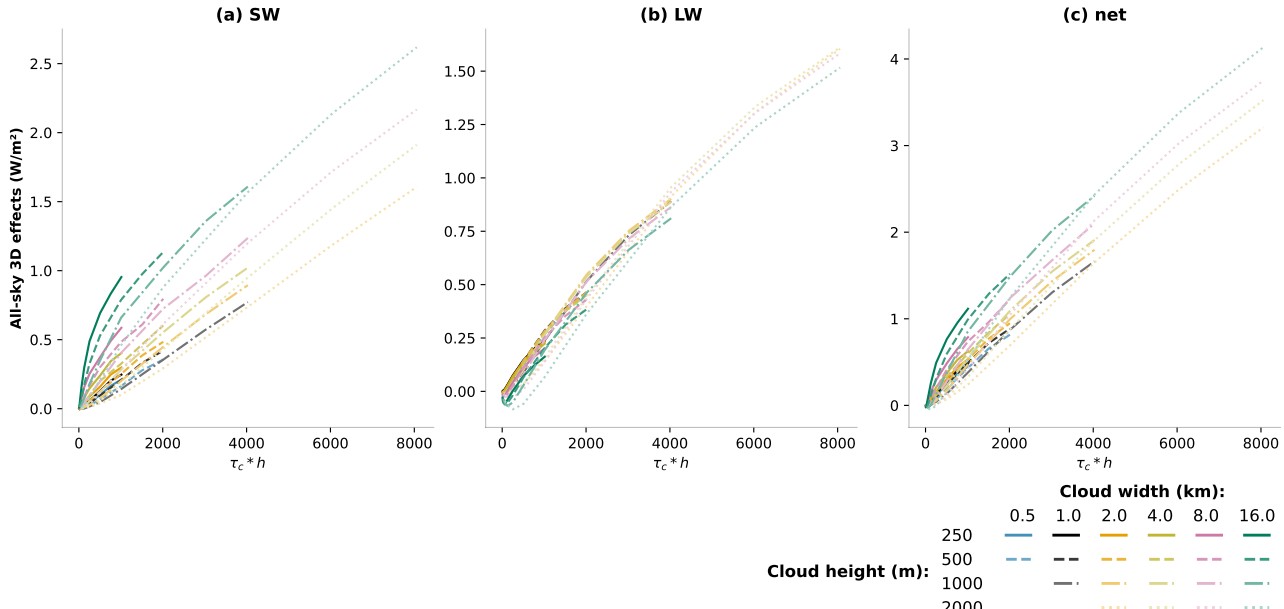

**Figure 5.** All-sky 3D effects as a function of the product of the total mass of ice, cloud width and cloud height.

We start with the analysis of plane parallel calculations. Figure 6(a) shows the SW oCRE as a function of SZA for several cloud optical depths, obtained with htrdr under parallel plane approximation (crosses) and with Eq. 3 (solid lines). Let's focus first on low optical depths, i.e. $\tau_c = 0.25$ (purple lines). In the SW, the cooling effect of the cloud strengthens with increasing SZA before reaching a maximum at large angles (70°-80°) and declining towards zero when the Sun gets close to the horizon (Fig. 6(a)). Two opposing effects are at play in the SW when the Sun goes down on the horizon: decreasing incident solar radiation, following the cosine of zenith angle $\mu_0$, and increasing cloud reflectivity $R(\mu_0)$.

The increase in cloud reflectivity can be analysed using MW80 (Eq. 3), which provides a good approximation of the dependence of reflectivity to SZA. This increase is driven by $\mu_0$ via two components; the first is the dependence of extinction on the path taken by incident solar radiation within the cloud (the exponential term in Eq. 3), the second is the angular dependence of scattering properties. The impact of these two components of cloud reflectivity can be distinguished by successively taking a constant $\mu_0 = 1$ in the exponential term of Eq. 3, and then setting $g = 0$ in the $\gamma_1$ and $\gamma_3$ parameters. Figure 6(b) displays the reflectivity as a function of SZA, and panel (c) shows the product of cloud reflectivity and cosine of the zenith angle, i.e. the combination of the two components of SW CRE. For optically thin clouds ($\tau_c = 0.25$), the $\mu_0 = 1$ curve (dashed line) shows a pronounced attenuation of the strong peak at large SZAs that exists using the unmodified MW80 expression (solid line) and obtained with htrdr. This difference can be attributed to the absence of increase in extinction resulting from setting a constant $\mu_0$. The remaining increase in cloud reflectivity in the $\mu_0 = 1$ curve can be attributed to the angular dependence of scattering effects, explained by the strong forward peak of ice crystals scattering phase function. When the Sun is at zenith, scattering directions towards the surface are favored. For increasing zenith angles, the probability of forward scattering is unchanged but





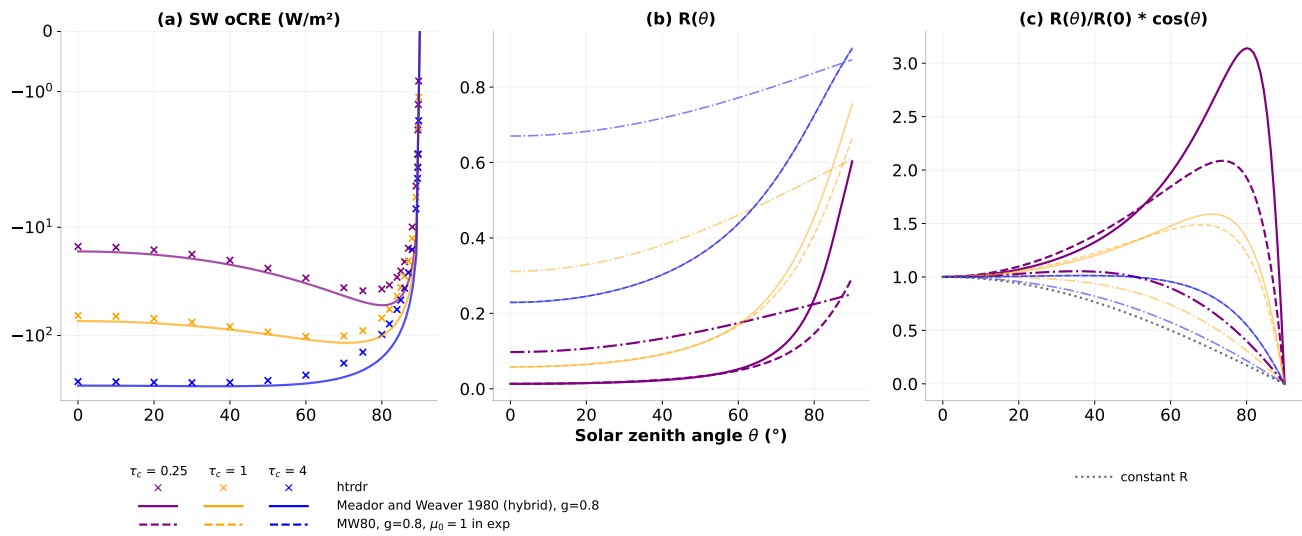

**Figure 6.** (a) Plane parallel computation of the overcast cloud radiative effect, in W/m$^2$, as a function of solar zenith angle $\theta$ for a cloud with width 1 km and height 500 m. Lines denote Meador and Weaver (1980) approximations and crosses denote htrdr calculations using the parallel plane method. Three cloud optical depths are shown : 0.25 in purple, 1 in yellow and 4 in blue. Note the logarithmic scale on the y-axis. (b) Cloud reflectivity and (c) Ratio of cloud reflectivity to cloud reflectivity at zenith multiplied by the cosine of solar zenith angle, for several approximations, as a function of solar zenith angle. Colors are as panel (a). Dashed lines show MW80 with $\mu_0 = 1$ in the exponential term, and dashdotted lines show MW80 with both $\mu_0 = 1$ and $g = 0$. The dotted grey line in panel (c) illustrates the case of a constant reflectivity, i.e. $\cos(\theta)$.

the scattering direction follows the SZA, ending up in a larger fraction of radiation being upscattered. This angular dependence of upscattering has been studied by Wiscombe and Grams (1976) among others, and can be highlighted by comparing to a computation where the asymmetry factor $g$ is set to 0. The resulting phase function is symmetrical and the angular dependence of upscattering is dampened. The obtained reflectivity, illustrated by the dashdotted lines in Fig. 6(b) and (c), clearly shows

the absence of the previously observed peak at large zenith angles. It is interesting to note in panel (b) that (1) as the $g = 0$ reflectivity has a weaker angular dependence, its value at zenith is larger than MW80 and (2) despite the higher value at zenith and a mild increase with increasing zenith angle, its value at 90 degrees is still lower than the MW80 reflectivity. For illustrative purposes, a constant reflectivity is added on panel (c) (dotted line). The $g = 0$ curve is close to the constant reflectivity curve, only small angular dependence remains, due to the $g = 0$ phase function not being perfectly isotropic and thus still having a

small angular dependence.

The maximum reached by the SW CRE at large SZA is specific to small cloud optical depths. As the cloud optically thickens, the reflectivity for the Sun at zenith increases and its angular dependence is diminished. As a result, the SW CRE reaches its maximum closer to zenith. The $\mu_0 = 1$ curve does not differ from the regular MW80 curve any longer. The increase with zenith




angle of the path taken by incident solar radiation within the cloud has no effect any more. The remaining observed angular

dependence of reflectivity is attributed to scattering properties only.

## 4.2    Variation of 3D effects with solar position

Three-dimensional effects of radiation influence how SW CRE depends on the position of the sun. Under plane parallel approximation, the azimuth angle has no influence on the SW CRE as the cloud is seen as infinite in both x and y horizontal directions. In 3D calculations however, the orientation of the Sun relative to the cloud needs to be accounted for. Figure 7

shows the 3D SW oCRE for two azimuth angles, corresponding to two orientations of the Sun: perpendicular ($\phi = 0$, dashed line) and parallel ($\phi = 90$, dotted line) to the cloud. Focusing first on low cloud optical depth, $\tau_c = 0.25$. For zenith angles from $\theta = 0°$ to about $\theta = 75°$, 3D effects are positive. Qualitative behavior when 3D radiative effects are included is similar to that for the 1D calculations: increase of SW CRE with increasing SZA, maximum located at large SZA and decrease to 0 at $\theta = 90°$. The location and magnitude of the maximum CRE, however, differ, and depend on azimuth angle. When the Sun

is low on the horizon and in a direction parallel to the cloud, 3D radiative effects tend to zero and 1D and 3D give almost the same results. In contrast, when the Sun is perpendicular to the cloud, the maximum of SW CRE is larger in the 3D case and happens at a larger SZA than the 1D case (around $\theta = 85°$), leading to negative 3D effects that can account for up to several hundred percents of the SW CRE. The divergence in 3D effects based on cloud orientation relative to the Sun can be interpreted with clarity. On one hand, the "parallel-Sun" 3D case is geometrically similar to the 1D approximation, with the Sun seeing

a contrail of infinite length parallel to its rays. In both scenarios, most of the radiation goes into the contrail and very little is scattered through the sides, yielding weak 3D effects. On the other hand, when the Sun is perpendicular to the contrail, and especially when it goes down on the horizon, an increasing fraction of cloud sides is sunlit, increasing its effective cover and resulting in enhanced 3D effects and SW CRE.

The zenith angle at which 3D effects change sign to become negative in the perpendicular configuration depends on the

cloud optical depth. It decreases from $\theta \sim 75°$ for $\tau_c = 0.25$, to $\theta \sim 45°$ for $\tau_c = 4$.

## 4.3    Sensitivity to cloud shape

To investigate the impact of our choice of a rectangular cross section for the cloud, experiment were repeated with an elliptical contrail. We chose a configuration close to the one studied in Gounou and Hogan (2007), hereafter GH07, with a width of 800 meters and height of 400 meters and mean optical depth of $0.2$ and $0.4$. The ice water content (IWC) depends on the distance

to the center of the contrail, with a peak IWC at the center, as in GH07:

$$IWC = \begin{cases} IWC_p \cos\left(\frac{\pi}{2}r\right) & \text{for } r < 1; \\ 0 & \text{for } r \geq 1, \end{cases} \tag{6}$$





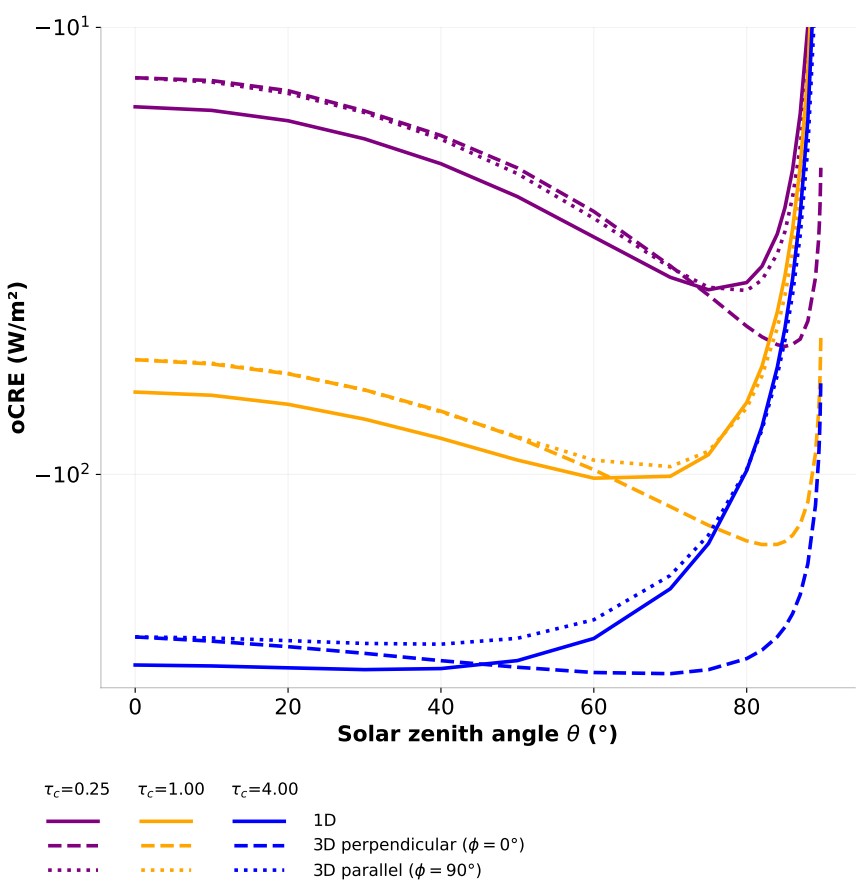

**Figure 7.** SW overcast cloud radiative effect as a function of solar zenith angle $\theta$. The solid lines denote htrdr calculations using the parallel plane approximation. The dashed and dotted lines correspond to 3D calculations, with an azimuth angle of $0°$ and $90°$, respectively. Three cloud optical depths are shown : 0.25 in purple, 1 in yellow and 4 in blue. The calculations are for the cloud with 1000 meters width and 500 meters height. Note the logarithmic scale on the y-axis.




where

$$r = \left[ \left( \frac{x - x_0}{\Delta x/2} \right)^2 + \left( \frac{z - z_0}{\Delta z/2} \right)^2 \right]^{\frac{1}{2}} \tag{7}$$

where $IWC_p$ is the maximum ice water content at the center of the cloud, $(x_0, z_0)$ the position of the center of the cloud and
$\Delta x$ and $\Delta z$ the geometrical width and thickness of the cloud. To separate the effects of IWC inhomogeneity and cloud shape, we also ran experiments with an ellipse of uniform IWC. We end up with 3 types of clouds with the same mean optical depth: the rectangle cloud with uniform IWC and optical depth, the uniform ellipse with uniform IWC and inhomogeneous optical depth, and the GH07 ellipse with inhomogeneous IWC and optical depth. The three configurations show different degrees of inhomogeneity in terms of IWC and cloud optical depth.

The CRE of the three configurations are very similar, both in 1D and 3D (Fig. 8). In the LW and in the SW at zenith, differences between the ellipse and the rectangular cloud CREs are very small. The LW oCRE of the GH07 ellipse is slightly weaker, i.e. less warming than that of the rectangular cloud, by less than $5\%$. The SW oCRE at zenith shows a small difference: the CRE of the GH07 ellipse exhibits a stronger cooling of about 0.5 W/m$^2$, i.e. $2.5\%$. At larger zenith angles, larger differences appear, where the rectangular cloud (uniform ellipse) is up to $15\%$ ($10\%$) more cooling than GH07 ellipse above 80 degrees.
For the net oCRE, this results in a GH07 ellipse less warming than the others at zenith and less cooling at large zenith angles where the net oCRE is negative. The delta oCRE changes sign at some point but around the same point the net oCRE also changes sign.

In terms of cloud optical depth inhomogeneity, differences are negligible in the LW and in the SW with the Sun at zenith, but for large SZAs, the SW oCRE increases as cloud optical depth becomes more homogeneous. Also, the differences between
uniform and non uniform clouds are more pronounced than between ellipses and rectangular clouds, which means that here, inhomogeneity seems to be more important than the cloud shape for the CRE, via optical depth effects (and therefore scattering) at the sides of the clouds.

But importantly, the three configurations show 3D effects of similar magnitude. Thus, the choice of a homogeneous rectangular cloud does not change the behavior of the CRE or 3D effects and even facilitates the analysis.

## 5 Temporal integration of the CRE and 3D effects

In the previous sections we calculated the sensitivity of the CRE to the Sun zenith and azimuth angles. In this section, we assess how CRE and 3D effects behave when integrated in time on a specific day and location. This allows a better representation of real life scenarios where all solar angles do not have equal weights. The usual method to calculate a daily mean is to fit the curve of the CRE as a function of the solar zenith and azimuth angles, followed by integrating the obtained curve over zenith
and azimuth angles in accordance to their distribution throughout the day. However, Monte Carlo methods offer another way, since they consist in statistical evaluation of integrals. The introduction of an additional integration dimension, in this case over the SZA, is equivalent to adding one random sampling per photon path, with minimal impact on numerical complexity. With this approach, time integration is performed without increasing computational time (Nyffenegger-Péré et al. (2024)). In





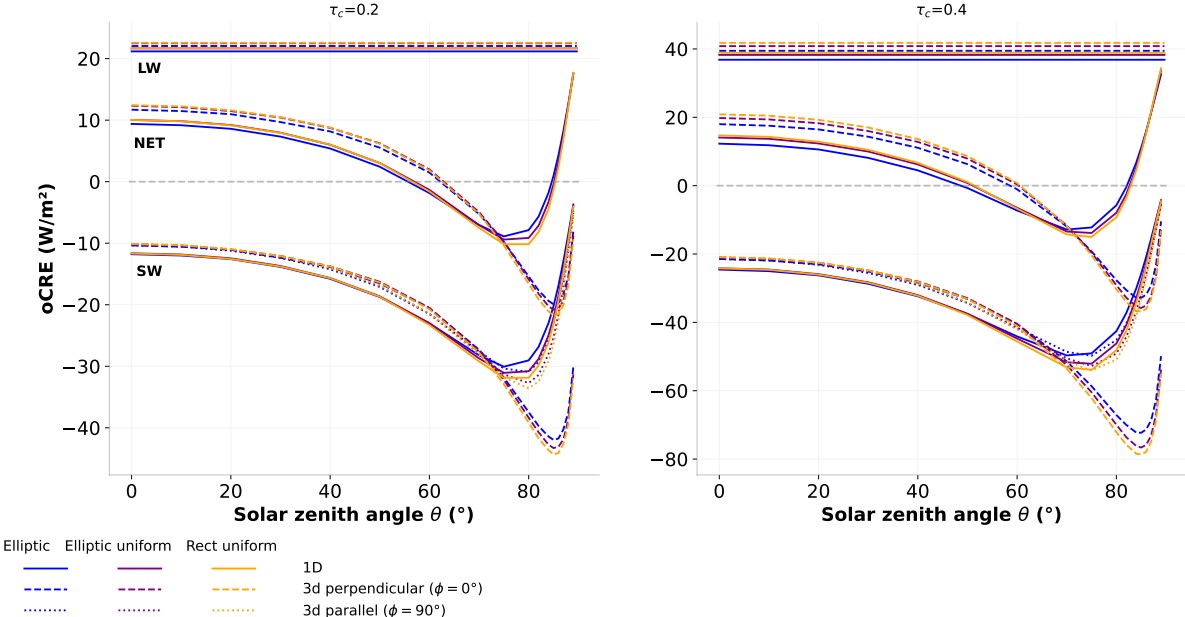

**Figure 8.** LW (top), SW (bottom) and net (center) oCRE (in W/m$^2$) as a function of solar zenith and azimuth angles, for a 800 m wide and 400 m thick cloud, for a cloud optical depth of $0.2$ (left panel) and $0.4$ (right panel). Clouds with elliptic shape and inhomogeneous IWC are in blue, elliptic shape with uniform IWC in purple, and rectangle shape with uniform IWC in orange.

order to achieve this, a solar position is sampled for each sampled photon, thus allowing the daily and spectral, spatial, angular,

photon-path integrations to be performed simultaneously. Daily averages of the CRE and 3D effects are calculated for a number of dates and locations. Two latitudes and 3 dates are considered, namely 45 and 60 degrees, equinoxes and winter and summer solstices.

Figure 9(a) shows the mean daytime CRE, i.e. the CRE averaged between sunrise and sunset, for the selected days and latitudes, and three cloud optical depths. As before, mean SW CRE increases with cloud optical depth. The impact of latitude

and day of the year varies depending on cloud optical depth: for the thinnest cloud ($\tau_c = 0.25$), SW CRE roughly increases with the day of the year and latitude (from summer solstice to equinox to winter solstice, and from 45 to 60 degrees of latitude), whereas in the cloud with optical depth of $4$, effects are more pronounced and in the opposite direction. This can be understood by examining the distribution of solar angles in each configuration and focusing first on the 1D calculations. As the day of the year progresses from summer solstice to equinox and then to winter solstice, solar zenith angles increase (Fig. D1). Winter

solstice especially stands out from the other days because its SZAs remain larger than $80°$, whereas only about 20% of SZAs are that low in the other configurations. Therefore, the mean daytime SW CREs over the year exhibit a similar pattern to that observed in Fig. 7: for the thinnest cloud with optical depth $0.25$, an initial increase in the SW CRE with zenith angle is followed by a decrease to zero after reaching a maximum at a high zenith angle around $80°$. For the thickest cloud with $\tau_c = 4$,





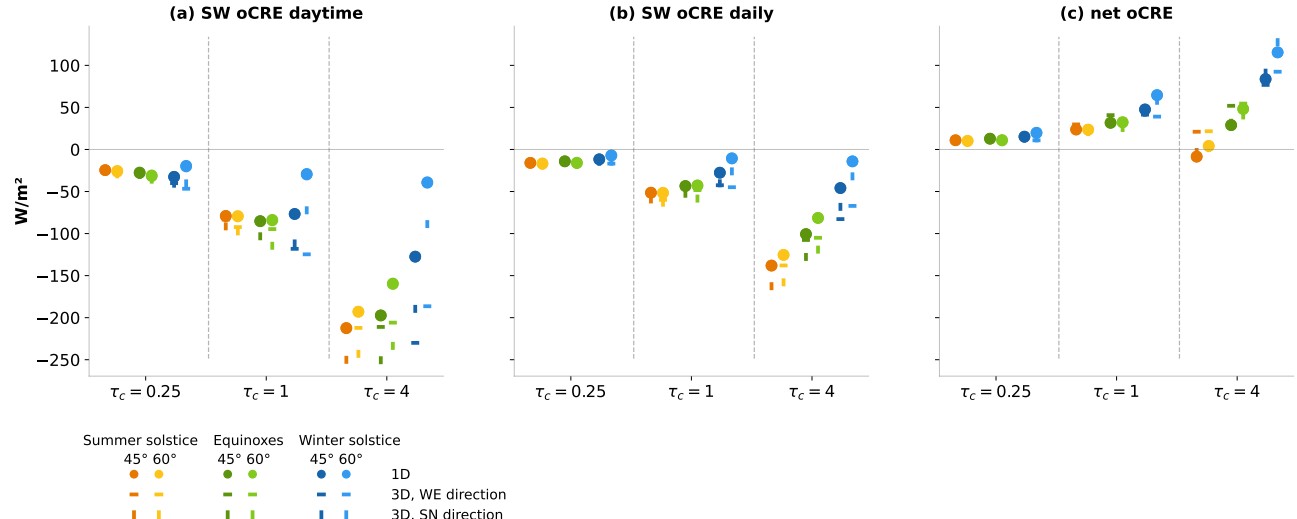

**Figure 9.** Mean (a) SW daytime, (b) SW daily and (c) net oCRE for selected days, latitudes, cloud optical depths and cloud orientations. Dots denote the 1D calculations, and dashes the 3D calculations: horizontal (vertical) dashes are for W-E (S-N) oriented clouds. In each panel are three groups of points corresponding to the three studied cloud optical depths. In each group, from left to right, is summer solstice, equinox and winter solstice, dark colors are for 45 degrees latitude and lighter colors for 60 degrees latitude. The calculations are for a rectangular cloud with 1000 meters width and 500 meters height.

Fig. 7 shows a maximum CRE at zenith before slowly decreasing to zero. As a consequence, the mean CRE of the $\tau_c = 4$ cloud
peaks in summer and decreases in the other days of the year according to the distribution of angles.

Daily mean SW CRE is obtained by multiplying daytime SW CRE by daytime fraction, as plotted in Fig. 9(b), having the effect of reducing the magnitude of the SW CREs, especially during winter solstice and equinoxes. Finally, the net daily effect can be calculated by adding up the LW CRE, which in the majority of cases yields a positive result 9(c).

We now focus on 3D effects plotted in Fig. 10, with the top row displaying absolute values and the bottom row showing
relative 3D effects. To improve figure clarity, we focus on clouds with optical depths of 0.25 and 1. In the majority of cases, daytime SW 3D effects are negative (Fig. 10(a)). They increase with increasing optical depth, day of the year (summer solstice to equinox to winter solstice) and latitude. Thus, largest 3D effects are observed at winter solstice at latitude 60 degrees. In addition to the dependence on cloud optical depth, day of the year and latitude, averaged 3D effects also exhibit a dependence on cloud orientation relative to the Sun. Here we examine again two configurations: one with the cloud aligned in a south-north
direction, and another in a west-east direction. As seen in section 4, 3D effects are influenced by both zenith and azimuth angles (Fig. 7): the largest 3D effects are observed at large zenith angles, when the Sun rays are perpendicular to the cloud. During the summer solstice and equinox, the perpendicular position is more prevalent for the S-N cloud, whereas for the winter solstice, it is more common for the W-E cloud. Given that the perpendicular orientation is associated with pronounced 3D effects at large zenith angles, the largest 3D effects are therefore caused by the S-N cloud during summer solstice and equinox, and by the



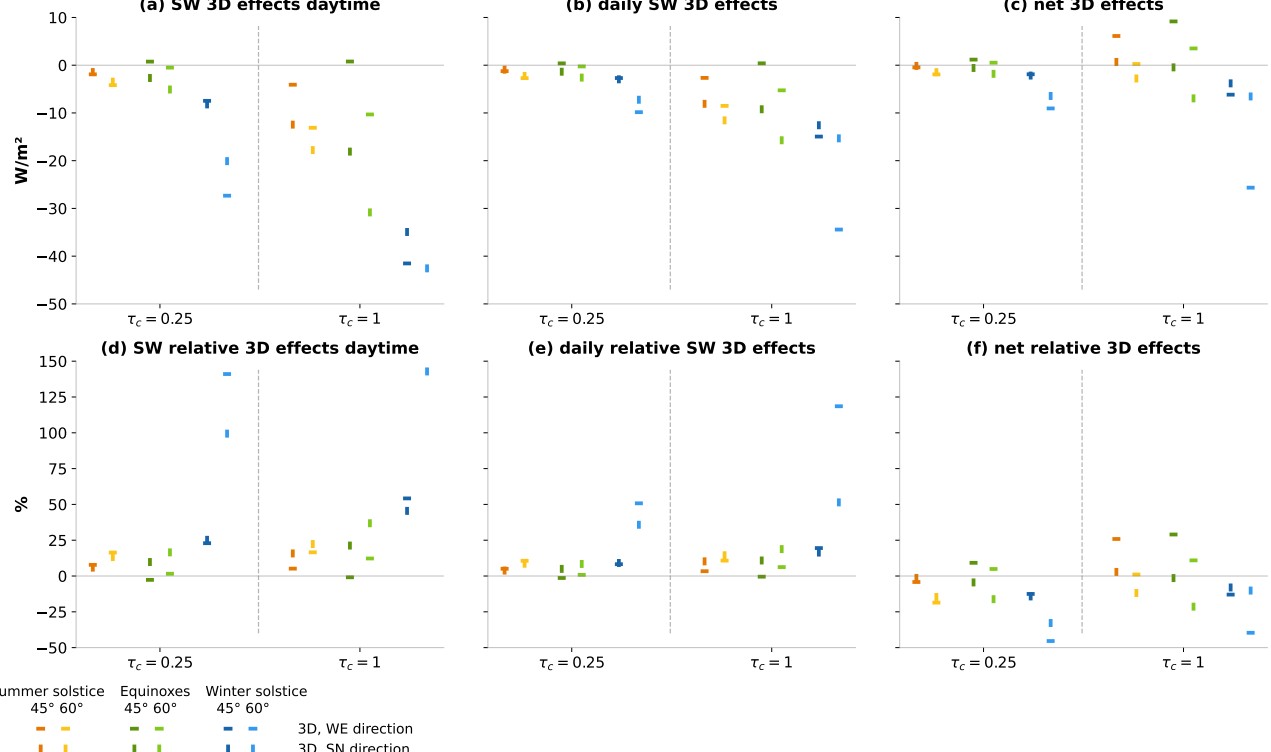

**Figure 10.** Mean (a) SW daytime, (b) SW daily and (c) net 3D effects and (d) SW daytime, (e) SW daily and (f) net relative 3D effects for selected days, latitudes, cloud optical depths and cloud orientations. Horizontal (vertical) dashes are for W-E (S-N) oriented clouds. In each group, from left to right, is summer solstice, equinox and winter solstice, dark colors for 45 degrees latitude and lighter colors for 60 degrees latitude. The scale on the y-axis has been adjusted to enhance the clarity of the figure; some extreme values have been excluded. The calculations are for the cloud with 1000 meters width and 500 meters height.

W-E cloud during winter solstice. This behaviour is evident in the $\tau_c = 1$ (and also for $\tau_c = 4$, not shown), and is also exhibited by the thinnest cloud, despite the fact that certain configurations do not show distinct 3D effects (for the summer solstice at both latitudes and the winter solstice at 45 degrees of latitude).

Similarly to the SW CRE, switching from daytime to daily SW 3D effects reduces their magnitude but does not change their sign (Fig. 10(b)). In contrast, adding the positive LW 3D effects to SW 3D effects to obtain net 3D effects yields positive values

in some cases (Fig. 10(c)). The highest relative 3D effects are observed in winter, 60 degrees of latitude. Even though they can go above $100\%$ of the CRE when considering only SW daytime 3D effects, net 3D effects remain under $50\%$ of the 1D CRE.





## 6  Conclusions

This paper investigates the sensitivity of the radiative effect of optically thin ice clouds to their geometric properties and optical depth, with a focus on three-dimensional effects of radiation and the subtle balance between SW and LW components of the net CRE. To this end, we use the Monte Carlo radiative transfer code htrdr on a parallelepipedic cloud with varying width, geometrical thickness and optical depth.

For sufficiently thin clouds and a Sun at zenith, the 1D SW all-sky CRE (aCRE), i.e. the radiative effect of the cloud with neighbouring clear sky parts, can be accurately determined using only the total ice content in a grid box. As long as the cloud optical depth is small, at least $\tau_c \leqslant 4$, cloud fraction does not influence the SW aCRE because for these optical depths, cloud reflectance is directly proportional to cloud optical depth, which only depends on ice water path. This is also valid for 1D LW aCRE but for a smaller range of optical depths ($\tau_c \leqslant 0.5$).

Simple analytical models reproduce the dependence of LW and SW oCRE on cloud optical depth quite well when taken separately. The LW CRE can be well approximated by a simple, classical model, whereas the SW CRE requires an adequate two-stream approximation, many of which, including the simple $(1-g)/2$ approximation, lead to large errors. When the LW and SW are added together to obtain the net CRE, however, small differences are exacerbated and the net CRE is less well reproduced.

The SW CRE varies significantly with zenith angle and displays different behaviors depending on optical depth. The CRE of very thin clouds reaches a maximum at large zenith angles ($\theta \sim 75°$ for $\tau_c = 0.25$), before decreasing to zero when the Sun goes below the horizon. For thickest optical depths, around 4, the CRE is maximum at zenith, follows a plateau, and then begins to decline around $\theta \sim 40°$. Cloud optical depth determines the zenith angle at which maximum SW CRE is reached: the optically thinner the cloud, the larger the zenith angle of maximum SW CRE.

Three dimensional effects of radiation are positive in the LW and depend linearly on the product $\tau_c \times \frac{h}{w}$, with $\frac{h}{w}$ the cloud aspect ratio, and can therefore be easily approximated for homogeneous clouds. In the SW, 3D effects are positive at zenith, and their behaviour at increasing zenith angles largely depends on the position of the Sun relative to the contrail. When the cloud is parallel to the Sun, SW 3D effects are maximum at zenith and decrease to zero with increasing zenith angle. Conversely, when the cloud is oriented perpendicular to the Sun, the SW 3D effects, which are positive at zenith, change sign to become negative and reach a maximum at a large zenith angle. The zenith angle at which SW 3D effects switch sign depends on the optical depth: the optically thinner the cloud, the greater the zenith angle from which the effect turns negative.

The behavior of 3D effects in relation to the position of the Sun naturally gives raise to the question of their value when integrated over the course of a day. Integrating the CRE and 3D effects on selected latitudes (45 and 60 degrees, where high-frequency flight routes and persistent contrail are currently located) and days of the year (equinoxes, solstices), we find that 3D effects resist to daily integration in some cases. The Sun-cloud absolute azimuth angle influences the magnitude of the mean 3D effects, the highest daytime SW 3D effects being in winter at 60 degrees latitude, for a cloud oriented in the West-East direction. For summer and equinoxes, the South-North orientation is associated with the highest 3D effects. When considering the net result, positive LW 3D effects that persist day and night help balance the strong negative daytime SW 3D effects. The



net 3D effects still end up being significant, particularly when a large fraction of the day is associated with large SZAs, i.e. during winter at high latitudes.

Our calculations differ from those of Gounou and Hogan (2007) (GH07) and Forster et al. (2012) (F12) in several aspects. There are slight differences between the LW and SW CREs at zenith, which are compensated for in the net CRE. The 3D effects of htrdr are less pronounced than those of GH07 and F12 in the LW. In the SW, however, htrdr produces larger 3D effects than GH07 and F12, particularly at large zenith angles. Overall, htrdr results are more similar to those of F12 than GH07.

We also ran the calculations using Fu (1996) and Fu et al. (1998) optical properties for a radius of 10 microns. The results are similar to what we find with Laurent Labonnote's LUT and are not shown in this paper. In general, we expect that changing the ice crystals shape will affect the magnitude of the CRE and the intensity of its maximum absolute value at large zenith angles via the asymmetry factor. This might also explain the slight differences between our results and those of GH07 and F12.

We have shown that zenith angle plays an important role on the CRE, both in 1D and 3D, especially when it becomes very large ($\theta > 85°$). However, Earth curvature and the solid angle of the Sun, which are usually neglected as we do here, may be non negligible in this range of solar angles, and would need to be taken into account for a correct estimation of the CRE, especially for days or flight trajectories that mostly span large SZAs. Furthermore, this requires the radiative code to accurately calculate scattering even when the phase function has a strong forward peak, which is not necessarily the case with codes developed to be very fast. These effects when the Sun is very low on the horizon are also very sensitive to the representation of the phase function. The use of Henyey Greenstein has limitation (Boucher (1998)) and might need to be revisited.

This paper does not address the radiative effects of heterogeneity of cloud properties, particularly ice content, beyond the very idealised case studied in Gounou and Hogan (2007) and Forster et al. (2012). Given the importance of non-linear effects of optical depth on the CRE, both in terms of angular dependence and three-dimensional effects, it is reasonable to conclude that such heterogeneities will influence the value of the CRE, therefore requiring a dedicated study.

Finally, in this study, the atmospheric configuration was significantly simplified, particularly with a non-reflecting surface and the absence of neighboring clouds. The tools we used are capable of handling more complex scenarios, and the Monte Carlo approach enables global-scale analysis. However, this will be the focus of a future study. Still, it is probable that 3D radiative effects of optically thin clouds are important in many situations, especially at low SZAs.

## Appendix A: Precisions on radiative transfer calculations

When performing radiative transfer calculations, great care must be taken to ensure that the radiative budget is closed by accounting for the radiative effect of all photons, irrespective of their direction of travel. For calculations at zenith, a large horizontal sensor of $L = 66$ km is sufficient to capture radiation, even radiation located far away from the cloud. When the direction of incoming photons from the Sun is no longer vertical, the fraction of photons going to the sides increases significantly. Vertical sensors placed near the cloud are then needed, denoted $S_2$ and $S_3$ and depicted in Fig. A1. This creates a "box" that ensure that photons scattered upward with a horizontal component are included in radiative flux calculations. In these situations, the length of horizontal sensor $S_1$ can be reduced.



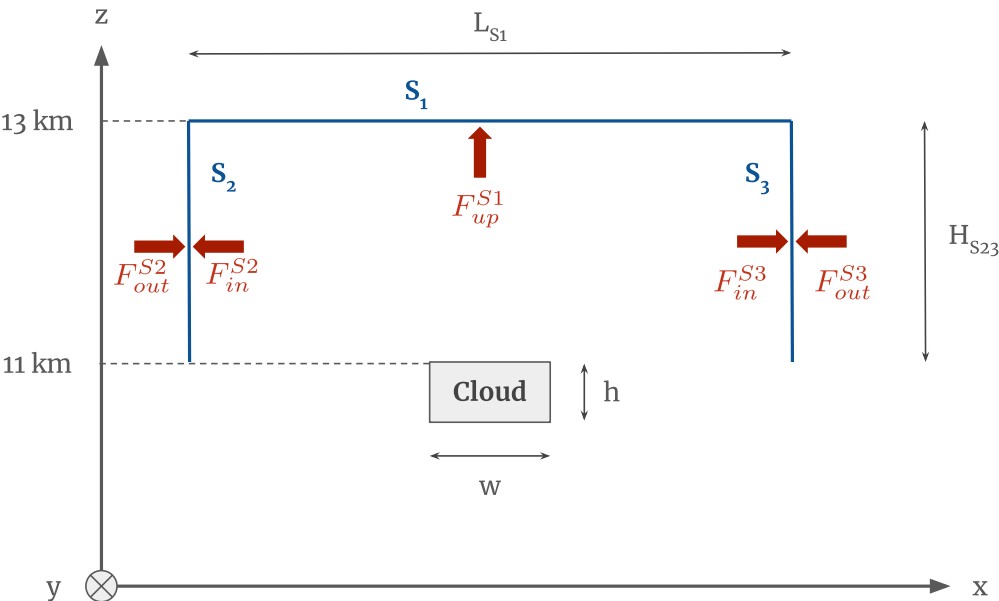

**Figure A1.** Schematic of the sensors used for the radiative transfer calculations. Sensor $S_1$ is used for all calculations when the Sun is at zenith, and sensors $S_2$ and $S_3$ are used when the solar zenith angle is larger than 0.

It also appeared that for the vertical sensors $S_2$ and $S_3$, the flux incident to the outer sides of the sensors (indirect flux, which does not come directly from the cloud) should be subtracted to the flux incoming to the inner sides, i.e. a net flux should be calculated for sensors $S_2$ and $S_3$. Not accounting for that inward flux causes errors in the magnitude, and even sign, of the integrated 3D effects for some geometries. Hence, 5 htrdr runs are needed to calculate a SW flux, and 10 for a SW CRE (5 clear, 5 cloudy). The 5 runs are added to give the corresponding total flux:

$$F = F_{up}^{S1} + \frac{H_{S23}}{L_{S1}} \left( F_{in}^{S2} - F_{out}^{S2} + F_{in}^{S3} - F_{out}^{S3} \right) \tag{A1}$$

with $w$ the cloud width, $L_{S1}$ the length of the horizontal sensor, $H_{S23}$ the height of the vertical sensors. For consistency, net fluxes at sensors $S_2$ and $S_3$ are multiplied by a factor $H_{S23}/L_{S1}$ to scale them to the width of sensor $S_1$. In the calculations in Section 4, $H_{S23}$ is set to 2 km, which is the distance between the altitudes of the cloud top and sensor $S_1$. $L_{S1}$ is set to 5 km. For plane parallel calculations, the second term on the right hand side of Eq. A1 is negligible since the size of the horizontal grid is multiplied by a very large number ($10^6$), thus giving $L_{S1} >> H_{S23}$.

The overcast CRE and 3D effects are then obtained with the following expressions:



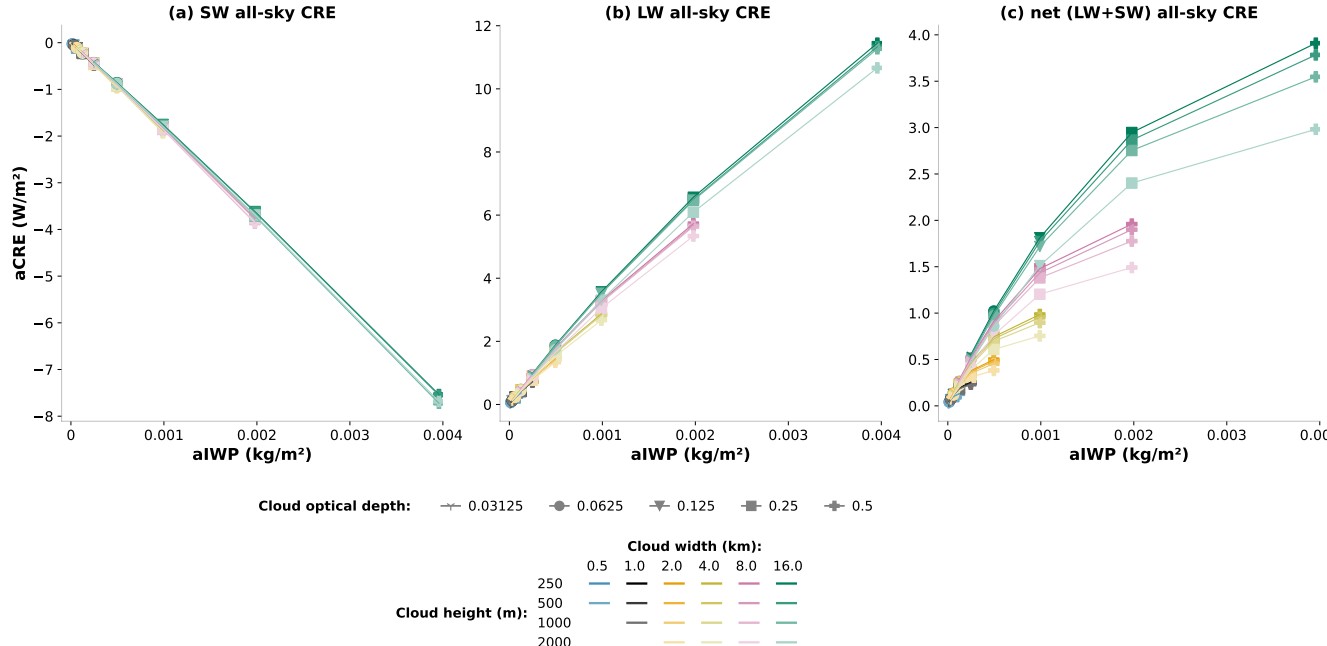

**Figure B1.** Same as Fig. 2, but zooming on cloud optical depths under 1 to highlight the linearity of LW CRE at these optical depths. SW CRE is also linear for this range of cloud optical depth and above.

$$oCRE = f \times aCRE = \frac{w}{L_{S1}} \left( F_{clear} - F_{all-sky} \right) \tag{A2}$$

$$3D = oCRE^{3D} - oCRE^{1D} = \frac{w}{L_{S1}} \left[ \left( F_{clear}^{3D} - F_{all-sky}^{3D} \right) - \left( F_{clear}^{1D} - F_{all-sky}^{1D} \right) \right] \tag{A3}$$

## Appendix B

Figure B1 is similar to Fig. 2, showing the aCRE of all cloud configurations as a function of the domain-averaged ice water
path (aIWP) for a Sun at zenith, except that it only shows configurations with a cloud optical depth below 0.5. This corresponds to the range of quasi-linearity between the aCRE and the aIWP in both spectral domains. As shown in section 3.1, the range of linearity for the SW extends to cloud optical depths up to 4.





## Appendix C: Simplified theoretical model

### C1 Details on the parameters used for MW80

We recall Eq. 3 for reflectivity:

$$R_{cloud} = \frac{1}{1+\gamma_1\tau_c}[\gamma_1\tau_c + (\gamma_3 - \gamma_1\mu_0)(1 - e^{-\tau_c/\mu_0})] \tag{C1}$$

Parameters $\gamma_1$ and $\gamma_3$ used in this expression come from Meador and Weaver (1980), using the modified Eddington-delta function hybrid method (also in Table C2).

$$\gamma_1 = \frac{7 - 3g^2 - \omega_0(4+3g) + \omega_0 g^2(4\beta_0+3g)}{4[1-g^2(1-\mu_0)]} \qquad \text{and} \qquad \gamma_3 = \beta_0 \tag{C2}$$

with $g$ the asymmetry parameter, $\omega_0$ the single scattering albedo, $\mu_0$ the cosine of solar zenith angle, and $\beta_0$ the backscatter function defined as:

$$\beta_0 = \frac{1}{2\omega_0}\int\limits_0^1 p(\mu_0, -\mu')d\mu' \tag{C3}$$

For values of the asymmetry parameter different than 0, we use the $\beta_0$ parameterized for the Henyey–Greenstein phase function by Barker and Li (1995), i.e.,

$$\beta_0 = \frac{16.156e^{-7.439g} + \mu_0[-0.148 + g(0.731 - 0.639g)]}{32.312e^{-7.439g} + \mu_0(4.357e^{-3.248g} + \mu_0)} \tag{C4}$$

and when $g = 0$, $\beta_0$ is set to $1/2$.

### C2 Previously published approximations of SW CRE

As mentioned in Section 3.1.1, we tested several approximations of the SW CRE found in the literature. These expressions are plotted in Fig. C1, with comparison to htrdr results. Most expressions are function of $g$, $\omega_0$, and $\tau_c$, except for the expression
from Schumann et al. (2012) which depends on the effective radius of ice crystals and 8 model parameters specific to the ice crystals shape. The values that we use for $g$, $\omega_0$ are listed in table C1.

Some approximations can be corrected : Meerkötter et al. (1999) has too strong a slope which can be corrected by applying a factor $(1-g^2)$. Pierrehumbert (2010) works better with a correction $(1-g)$ in term $\beta$, i.e. applied to the cloud optical depth in the exponential term. It is interesting to note that Pierrehumbert (2010) specifies that the factor $(1-g)$ should not be applied in
the $\beta$ term, despite the formula fitting better when applying it. The common $(1-g)/2$ used in an extensive number of studies in the case of thin clouds differs from htrdr. Surprinsingly, the very simple expression $(1-g)/4$ is one of the closest to htrdr. However, this might be a coincidence and we cannot conclude that this expression fits in every case.



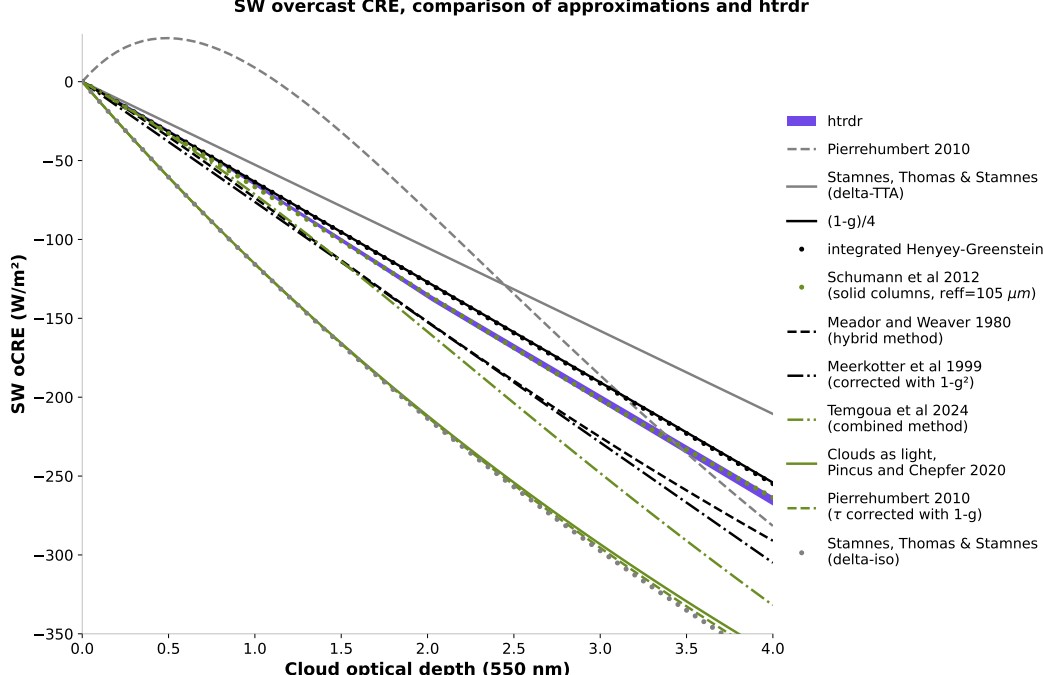

**Figure C1.** Comparison between SW CRE calculated with htrdr with the plane parallel approximation (purple) and the different expressions found in the literature listed in Table C2, for a Sun at zenith, as a function of cloud optical depth.

Schumann et al. (2012) model does not use the same parameters but fits htrdr remarkably well when used with single columns for the ice crystal shape and an effective ice crystal radius of 100 microns. This is expected, as Schumann's model parameters have been derived from a large set of radiative transfer computations using RT code libRadtran (Mayer and Kylling (2005)) with DISORT 2.0 solver (Stamnes et al. (1988)). The curve labelled "integrated Henyey Greenstein" on Figure C1 is obtained by integrating the Henyey-Greenstein scattering phase function over backscattering angles (between $\mu = -1$ and $\mu = 0$). This only works for a Sun at zenith where the upscatter fraction and the backscatter fraction are equal. It is expected to be in good agreement with htrdr since the scattering phase function implemented in the model is Henyey-Greenstein.



| LW | |
|---|---|
| $T^*$ (K) | 270 |
| $T_{surf}$ (K) | 300 |
| $\omega_0$ | 0.6 |
| $\bar{\mu}$ | 0.6 |
| SW | |
| $\omega_0$ | 1 |
| $\mu_0$ | 1 |
| g | 0.8 |
| $F^{\downarrow}$ (W/m$^2$) | 1270 |

**Table C1.** Parameters used in calculations with the LW and SW CRE expressions listed in Table C2 and shown on Fig. C1.

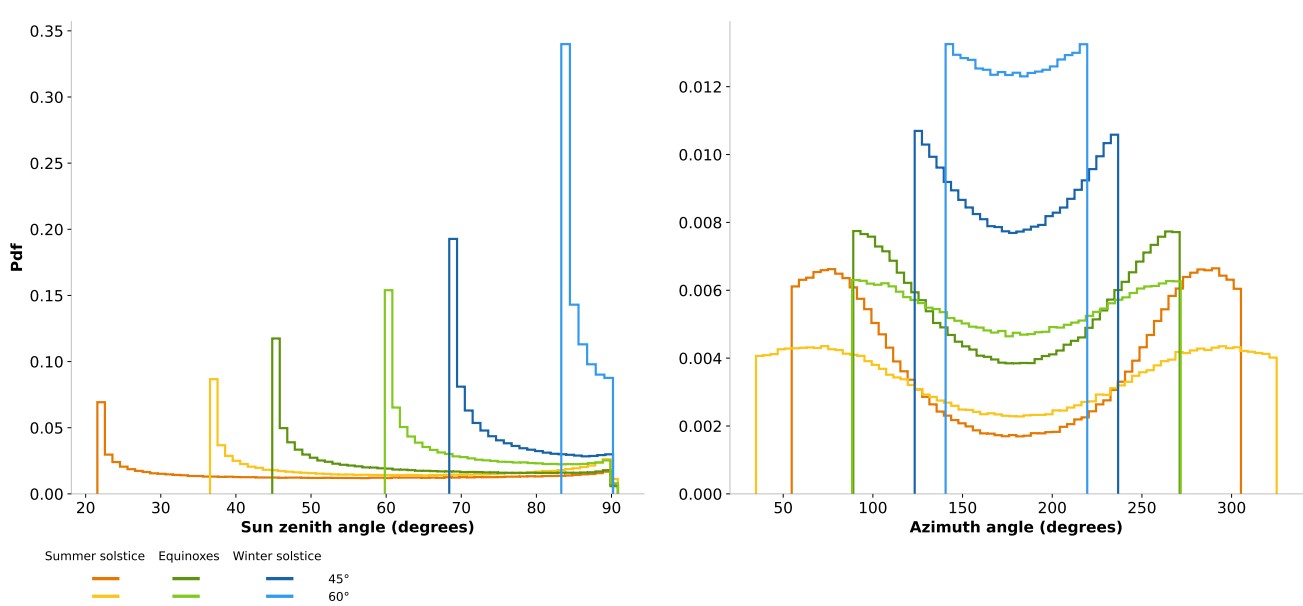

**Figure D1.** Histogram of Sun zenith (left) and azimuth (right) angles for selected days of the year and latitudes. The displayed angles correspond to the daytime period of the days.

**Appendix D: Histograms of solar angles**

Figure D1 shows histograms of zenith and azimuth angles for the 3 selected days of the year, for 45 and 60 degrees of latitude. The figure illustrates the stark contrast in the distribution of solar angles between summer and winter. At 60 degrees of latitude in the winter solstice, zenith angles are concentrated in ranges above 80 degrees. Given the challenges associated with calculating fluxes for angles within this range, this a subject requires further investigation.

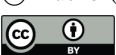

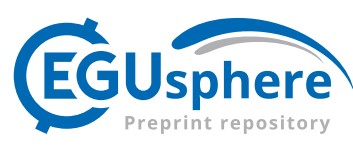

| Reference | Expression for reflectivity | Calculations of parameters |
|---|---|---|
| Meador and Weaver (1980) | $\dfrac{\omega}{1+\gamma_1\tau}\left(\tau\gamma_1 + (\gamma_3 - \mu_0\gamma_1)(1 - e^{-\tau/\mu_0})\right)$ | $\gamma_1 = \dfrac{7 - 3g^2 - \omega_0(4+3g) + \omega_0 g^2(4+\beta_0+3g)}{4(1-g^2(1-\mu_0))}$ and $\gamma_3 = \beta_0$ |
| Schumann et al. (2012) | $t_A^2\left(1 - e^{-\Gamma\tau'/\mu_0}\right)\left(C_\mu + A_\mu e^{-\gamma\tau'}\left(\dfrac{(1-\mu_0)^{B_\mu}}{(1/2)^{B_\mu}} - 1\right)\right)$ | $\tau' = \tau(1 - Fr(1 - e^{-\delta_{sr}r_{\mathrm{eff}}})),\ r_{\mathrm{eff}} = 105\ \mu m$, and the 10 model parameters for solid columns |
| Pierrehumbert (2010) | $\dfrac{(\frac{1}{2} - \gamma\mu_0)(1 - e^{-\tau/\mu_0}) + (1-g)\gamma\tau}{1 + (1-g)\gamma\tau}$ | $\gamma = 3/4$ (Eddington approximation) and corrected with $(1-g)$ in the exponential. |
| Stamnes et al. (2017) (page 252, delta - TTA) | $\tau_c(1-g)\frac{1}{2}(1 + g(1 - 3\bar\mu\mu_0))$ | $\bar\mu = 1/\sqrt{3}$ |
| Stamnes et al. (2017) (page 262, delta - iso) | $\dfrac{2b\tau + (\bar\mu - \mu_0)(1 - e^{-2b\tau/\mu_0})}{2b\tau + 2\bar\mu}$ | $b = (1-g)/2$ and $\bar\mu = 1/2$ |
| Meerkötter et al. (1999) Integrated HG | $\left(\frac{1}{2} - \frac{3}{4}\frac{g}{1+g}\mu_0\right)\frac{\tau_c}{\mu_0}$ $\tau_c \displaystyle\int_{-1}^{0} \frac{1-g^2}{(1+g^2-2g\mu)^{2/3}}d\mu$ | Corrected with a factor $(1-g^2)$ |
| Temgoua et al. (2024) | $\dfrac{\omega}{1+\gamma_1\tau}\left(\tau\gamma_1 + (\gamma_3 - \mu_0\gamma_1)(1 - e^{-\tau/\mu_0})\right)$ | $\gamma_1 = \dfrac{1}{4(2g^2\mu_0 + (1-g^2)\tau)}\{7\tau + (8 - 7\tau)g^2 - \omega(4+3g)\tau + g^2\omega(4(\tau-1) - 4g^2 - 6g\mu_0 + 6g^2\mu_0 + 3g\tau)\}$ and $\gamma_3 = \beta_0$ |
| Siebesma et al. (2020) | $\dfrac{(1-g)\tau_c}{2 + (1-g)\tau_c}$ | |

**Table C2.** List of expressions used to calculate cloud reflectivity at zenith in Fig. C1. Values of the parameters common to all expressions are given in Table C1.




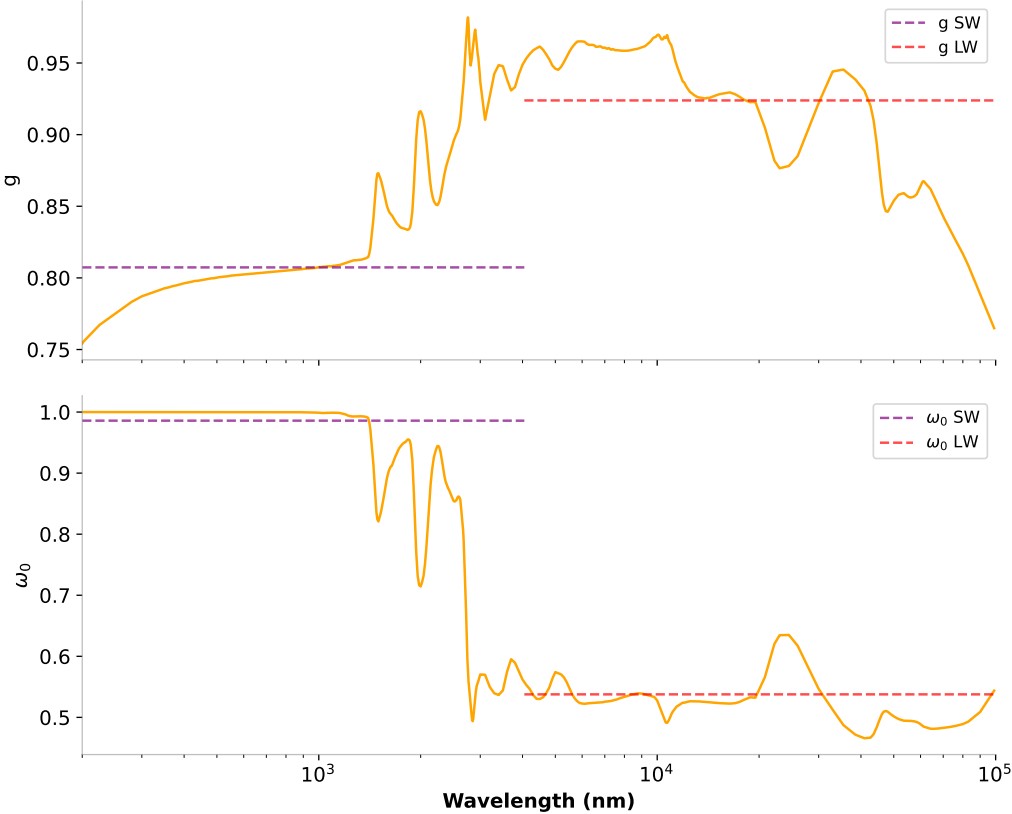

**Figure E1.** Asymmetry parameter (top) and single scattering albedo (bottom) of ice crystals as a function of wavelength, and their averages over the SW (purple) and LW (red) spectral domains in dashed lines. Note the log scale on the x-axis.

## 515  Appendix E:  Optical properties of ice crystals

Figure E1 shows the wavelength-dependent asymmetry parameter and single scattering albedo of ice crystals used for the radiative transfer calculations in this study. It is interesting to calculate their average over the SW and LW spectral domains, weighed by the Planck function at the Sun temperature ($T = 6000$ K) for the SW and cloud temperature ($T = 229$ K) for the LW, represented in dashed lines in Fig. E1. The spectral range of integration is between 0 and 4000 nm in the SW and 4000

and 100000 nm in the LW. The average values are very close to the ones used in the simple model described in the paper, and consistent with values used in other studies for cirrus and contrails (Myhre et al. (2009)).

*Author contributions.*  JC, JLD and NB conceptualized the paper. JC carried out the analysis and designed the figures. JC, JLD and NB wrote the original manuscript. NV brought helpful comments as well as useful help with htrdr. JC, JLD, NB and NV reviewed and edited the paper. All authors have read and agreed to the final version of the paper.



*Competing interests.*  The authors declare that they have no conflict of interest.

*Acknowledgements.*  JC and NB acknowledge the support of the French Ministère de la Transition écologique et solidaire (N° DGAC 382 N2021-39), with support from France's Plan National de Relance et de Résilience (PNRR) and the European Union's NextGenerationEU, through Climaviation research action. The authors would like to thank the ESPRI mesocentre at IPSL for access to computing resources, and Vincent Forest for his support with htrdr.



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
