# Peer review of "The subtleties of three-dimensional radiative effects in contrails and cirrus clouds"

_EGUsphere, 2024_

## Author Comment (AC1)

**Reply to the reviewers comments on the manuscript "The subtleties of three-dimensional radiative effects in contrails and cirrus clouds" by Carles et al.**

The authors would like to thank the reviewers for reading the manuscript and for their insightful comments that helped improve it. The following document provides answers to the comments in blue.

The line numbers given in this document correspond to the latest submitted version. All changes brought to the new manuscript version are noted in the track changes file provided.

Response to reviewer #1, comment RC1 :

The manuscript investigates cloud radiative effect (CRE) (both overcast and all sky) and 3D radiative effect using a Monte Carlo ray tracer run in a plane parallel (1D) and 3D configuration. It investigates the CRE (both shortwave and longwave) of contrails/cirrus considering a) different idealized geometries, b) for 1D vs. 3D calculations, and c) varying solar angles (including time-integrated solar angles). It also considers the 3D radiative effects of contrails for a) the sun at zenith with varying geometries and b) varying solar angles (including time integrated angles) with set geometry. These many analyses are accompanied by thoughtful discussion.

The manuscript contains several neat results:

- Overcast 3D radiative effects for the sun at zenith are positive (warming) for both the SW and LW due to side leakage and side cooling, respectively.
- When varying SZA, 3D radiative effect in the SW switches in sign from positive to negative at high solar zenith angle with a dependence on cloud optical thickness.
- In looking at time integrated solar position and effect on 3D radiative effect, there is a strong dependence on the orientation of the cloud relative to the sun and overall the strongest 3D effects are seen for the winter solstice.

This thorough manuscript would most benefit from better framing. What questions motivated the study? What have the authors learned that was not previously known or anticipated? How might the results be used to generalize understanding of how contrails affect radiation?

We thank the reviewer for these comments and questions. We reformulated and added framing in the introduction, l. 28 to 36, (l. 31 to 39 in the track changes file): "The three-dimensional effects of radiation are not taken into account in contrail studies based on climate modelling (Bickel et al., 2020) or simplified process-models (Teoh et al., 2020, 2024). However, they have previously been demonstrated to be non negligible in the radiative effect of contrails and cirrus (Gounou and Hogan, 2007; Forster et al., 2012). These few pioneering studies are not yet sufficient to quantify 3D effects on a global scale, e.g. to know in which situations they are most significant, whether they always have the same sign, or whether they could be at least partially compensated when considering temporal (diurnal, seasonal cycle) or spatial averages (along a flight trajectory). Thus, the

objective of our study is twofold : firstly, to quantify the 3D effects of radiation in contrails and identify the behavior specific to a range of low optical depths representative of these clouds and secondly to explore the importance of the 3D effects when integrated on idealised larger time or space scales".

See also the conclusion, l. 427 to 429 (l.455 to 457 in the track changes file): "The identification of configurations in which 3D effects are most important paves the way for the validation of existing parameterizations of 3D effects (Schäfer et al., 2016; Hogan et al., 2016), or the development of new parameterizations adapted to these particular situations."

The manuscript would also benefit from greater emphasis on novel insights. The manuscript reports on both 1D and 3D calculations of CRE. The former are less novel than the latter, and the new results might come out more clearly if the 1D results were reported more selectively. In particular, the linear relationship between ice water path and CRE for the low cloud optical thickness regime is well established in the literature (and summarised in the authors' table C2); the manuscript could instead rely upon citations to allow for more attention on 3D radiative effects.

We have re-organized section 3.1.1 on 1D SW CRE and moved some text to Appendix C1. This has allowed us to shorten this section to put more emphasis on the 3D sections. Always with the aim of shortening 1D sections, we have removed a paragraph in section 3.1.3 on 1D net CRE (l. 220 to 226 in the track changes file).

Some more specific suggestions:

● Figures 9 and 10 were the most interesting to me but would benefit from a diagram of the N-S, W-E clouds and path of the sun at equinox and solstice to further illustrate the author's insights.

We added a figure of sun paths, Fig. 9, to help vizualisation.

● It would be easier to compare between sections if there were consistency in language about cloud orientation; the authors use parallel, perpendicular in figure 8 and N-S, E-W in figures 9 and 10.

We understand the reviewers request for more consistency, but the cloud orientations in the two sections are not strictly comparable. In section 5, the terms "parallel" and "perpendicular" cannot be used because the position of the sun relative to the cloud changes during the day. However, the new figure of sun paths, Fig. 9, may provide a better understanding of the configurations.

● Figure 7 would benefit from a second panel showing the 3D radiative effects so the reader does not have to do the calculation with their eye.

Following the reviewer's suggestion, we added panels of SW 3D effects and 3D effects relative to 1D calculations in Fig. 7. This led us to also add a panel of relative 3D effects on Fig. 8 for the sake of comparison between Fig. 7 and 8.

● While the scope of the study does not include the interplay between geometry and solar angles on 3D radiative effects, synthesis and discussion on this topic would be very interesting.

We thank the reviewer for this comment. We added a paragraph in the conclusion, l. 430 to 434 (l.458 to 462 in the track changes file), about this topic : "The effect of solar position on SW 3D effects has been studied for a cloud with a fixed aspect ratio of 1/2 (a cloud width of 1 km and geometrical thickness of 0.5 km). We expect the SW 3D effects to go to decrease as the aspect ratio goes to zero, but not linearly. Calculations conducted on a cloud with an aspect ratio of 1/8, i.e. a cloud four times larger (width of 4 km and thickness of 0.5 km, with a cloud optical depth of 0.25), show that the daytime integrated SW 3D effects do not decrease by the same amount. This suggests that 3D effects of radiation may still be significant for clouds with small aspect ratios."

---

## Author Comment (AC2)

**Reply to the reviewers comments on the manuscript "The subtleties of three-dimensional radiative effects in contrails and cirrus clouds" by Carles et al.**

The authors would like to thank the reviewers for reading the manuscript and for their insightful comments that helped improve it. The following document provides answers to the comments in blue.

The line numbers given in this document correspond to the latest submitted version. All changes brought to the new manuscript version are noted in the track changes file provided.

Response to reviewer #2, comment RC2 :

General comments

The study by Carles et al. is a sensitivity analysis of optically thin ice clouds radiative effects (CRE) to their geometrical dimensions and optical thickness investigating also the importance of the 3D effects on these estimations. By using Monte Carlo radiative transfer simulations for both 1D and 3D configurations, the authors derived useful insights for CREs (SW, LW and net) in terms of clouds geometry, their optical thickness and the relative position of the sun, highlighting when the 3D effects are important. To put into perspective their results in terms of the importance of including the 3D effects in estimating cirrus and contrails radiative effects and forcing, CRE estimates integrated for selected days at different latitudes were also calculated and discussed. The objectives of the study are quite straightforward and are addressed by thorough analysis. I consider the topic and results of this manuscript to fit the scope of ACP.

I have some general and minor comments which should be addressed prior to publication.

● It would increase the value of manuscript to elaborate in the introduction which was the gaps identified in previous studies which motivated the objectives of the current study.

We thank the reviewer for this comment. We reformulated and added framing in the introduction, l. 28 to 36, (l. 31 to 39 in the track changes file): "The three-dimensional effects of radiation are not taken into account in contrail studies based on climate modelling (Bickel et al., 2020) or simplified process-models (Teoh et al., 2020, 2024). However, they have previously been demonstrated to be non negligible in the radiative effect of contrails and cirrus (Gounou and Hogan, 2007; Forster et al., 2012). These few pioneering studies are not yet sufficient to quantify 3D effects on a global scale, e.g. to know in which situations they are most significant, whether they always have the same sign, or whether they could be at least partially compensated when considering temporal (diurnal, seasonal cycle) or spatial averages (along a flight trajectory). Thus, the objective of our study is twofold : firstly, to quantify the 3D effects of radiation in contrails and identify the behavior specific to a range of low optical depths representative of these clouds and secondly to explore the importance of the 3D effects when integrated on idealised larger time or space scales".

● There is an extensive part dedicated to CRE 1D results and to simple analytical models of CRE in relation to cloud optical depth. Apart from explaining the results in a more intuitive way, please provide the added values of this analysis.

Following this comment and a similar comment from reviewer #1, we have re-organized section 3.1.1 on 1D CRE and moved some text to Appendix C1. This has allowed us to shorten this section and put more emphasis on the 3D sections. Always with the aim of shortening 1D sections, we have removed a paragraph in section 3.1.3 on 1D net CRE (l. 220 to 226 in the track changes file). See also the conclusion, l. 401 to 404 (l. 428 to 431 in the track changes file).

● There is a very short discussion in Lines 428-431 comparing the results of the present study with previous studies. I suggest placing at the relevant sections the differences and similarities that are briefly mentioned in the conclusions section.

We thank the reviewer for this comment. We decided not to move this paragraph, as no pre existing section was thought to be relevant (for instance, section 4.3 is about sensitivity to cloud shape and not a comparison between previous studies and our results), and we did not want to create a new section for this short discussion.

Specific comments

Table 1. L hasn't been introduced yet.

We corrected this in the table 1.

Line 154: I cannot see the vertical line mentioned here in Fig. 2b

We corrected Fig. 2.

Line 168: Please, be more specific instead of "cf"

Corrected in the text l.173 (l. 185 in the track changes file)

Figure 4: Fig 4b should have the colored light shadowing? I was expecting something more like Fig. 4c. In addition, "Blue" are black lines right?

We precised the legend: the shadowings in panel (b) correspond to the range of 3D effects. We also corrected "blue" by "black".

Technical corrections

Figure 3: (a) in the SW, (b) in the LW instead of "(a) in the LW, (b) in the SW"

We thank the reviewer for their attention, we have corrected the legend.

---

## Author Response (AR2)

Reply to the reviewer's comment on the manuscript "The subtleties of three-dimensional radiative effects in contrails and cirrus clouds" by Carles et al.

In black is the reviewer's comment, in blue the authors' response.

I admit the improvements on the revised manuscript. The manuscript is under the scope of the journal. I recommend to accept the paper after the following minor corrections:

1. Through the revised manuscript, independant should be independ'e'nt. Furthermore, the independent column approximation is already abbreviated as 'ICA' in the manuscript; I suggest to unify the word through the manuscript.

We thank the reviewer for the typos, we corrected them through all the manuscript and unified the "ICA" notation.

- 2. The previous studies (e..g., Stephens et al., 1991; Cahalan et al., 1994; Marshak et al., 1995; Varnai and Davies, 1999; Wissmeier et al., 2013; Okata et al., 2017) proposed several approximations based on the independent pixel/column approximation. The reviewer suggests for the authors to discuss among the methods in introduction. We added a sentence and reformulated the paragraph in the introduction, I.29 to 34.
- 3. L.296: 'Under plane parallelthe independant column approximation' should be 'Under the independent column approximation'

We thank the reviewer for their attention and corrected the sentence.

4. L.446: It is not fully correct because Momoi et al. (2022) proposed the method to take forward-scattering into account correctly yet rapidly.

We thank the reviewer for the reference, we added it I.452.

**Reference(s)**

Cahalan, R. F., S. Gollmer, W. J. Wiscombe, W. Ridgway, and Harshvardhan (1994), Independent pixel and monte carlo estimates of stratocumulus albedo, J. Atmos. Sci., 51(24), 3776–3790, doi:10.1175/1520-0469(1994)051<3776:IPAMCE>2.0.CO;2.

Marshak, A., A. Davis, W. Wiscombe, and R. Cahalan (1995), Radiative smoothing in fractal clouds, J. Geophys. Res., 100(D12), 26247, doi:10.1029/95JD02895.

Momoi, M., Nakajima, T., Irie, H., Okata, M., 2022. Efficient calculation of radiative intensity in three-dimensional atmospheres based on the pn-IMS truncation method with a backward ray tracing system. J. Quant. Spectrosc. Radiat. Transf. 293, 108369. https://doi.org/10.1016/j.jqsrt.2022.108369.

Okata M, Nakajima T, Suzuki K, Inoue T, Nakajima TY, Okamoto H. A study on radiative transfer effects in 3-D cloudy atmosphere using satellite data. J Geophys Res Atmos 2017;122:443–68. doi: 10.1002/2016JD025441.

Stephens, G., P. Gabriel, and S. Tsay (1991), Statistical radiative transport in one-dimensional media and its application to the terrestrial atmosphere, Transp. Theory Stat. Phys., 20(2), 139–175.

Varnai, T., and R. Davies (1999), Effects of cloud heterogeneities on shortwave radiation: Comparison of cloud-top variability and internal heterogeneity, J. Atmos. Sci., 56(24), 4206.

Wissmeier, U., R. Buras, and B. Mayer (2013), paNTICA: A fast 3D radiative transfer scheme to calculate surface solar irradiance for NWP and LES models, J. Appl. Meteorol. Climatol., 52(8), 1698.